# Parameterizing Non-Parametric Meta-Reinforcement Learning Tasks via Subtask Decomposition

**Suyoung Lee,    Myungsik Cho,    Youngchul Sung**[*]
School of Electrical Engineering
KAIST
Daejeon 34141, Republic of Korea
{suyoung.l, ms.cho, ycsung}@kaist.ac.kr

## Abstract

Meta-reinforcement learning (meta-RL) techniques have demonstrated remarkable success in generalizing deep reinforcement learning across a range of tasks. Nevertheless, these methods often struggle to generalize beyond tasks with parametric variations. To overcome this challenge, we propose Subtask Decomposition and Virtual Training (SDVT), a novel meta-RL approach that decomposes each non-parametric task into a collection of elementary subtasks and parameterizes the task based on its decomposition. We employ a Gaussian mixture VAE to meta-learn the decomposition process, enabling the agent to reuse policies acquired from common subtasks. Additionally, we propose a virtual training procedure, specifically designed for non-parametric task variability, which generates hypothetical subtask compositions, thereby enhancing generalization to previously unseen subtask compositions. Our method significantly improves performance on the Meta-World ML-10 and ML-45 benchmarks, surpassing current state-of-the-art techniques.

## 1  Introduction

Meta-reinforcement learning (meta-RL) constitutes a dynamic field within deep reinforcement learning, focusing on training agents to quickly adapt to novel tasks by learning from a variety of training tasks [2]. By interacting with these tasks, meta-RL creates an inductive bias regarding the task dynamics and subsequently develops a policy based on this knowledge. Despite its significant contribution to the generalization capability of traditional deep RL, meta-RL is susceptible to test-time distribution shifts, which restricts its applicability to familiar in-distribution test tasks [17, 36, 39, 41].

To tackle this limitation, recent out-of-distribution (OOD) meta-RL approaches have emphasized distinct training and test task distributions, thereby achieving enhanced performance on unseen OOD test tasks with interpolated or slightly extrapolated training dynamics [12, 36, 41, 35, 1]. Although the parameters of training and test tasks are drawn from disjoint distributions, these tasks remain qualitatively similar, as they can be expressed in a shared parametric form representing the task dynamics (e.g., the same "Pick-place" task with OOD goal positions in Figure 1b).

In this study, we explore a more general meta-RL framework that addresses non-parametric task variability [69, 71], a topic that has received limited attention in prior research. In this context as of Figure 1c, variations among tasks cannot be expressed through simple parametric variations, such as the parameterization of a goal position. Generalization is particularly challenging in this setting, as conventional meta-RL methods often model the inductive bias as a parametric embedding applicable to various tasks [9, 46, 74]. Within a non-parametric framework, it may not be feasible to employ a unified and generalizable parameterization of training tasks using standard meta-RL techniques.

---

[*]Corresponding author

37th Conference on Neural Information Processing Systems (NeurIPS 2023).

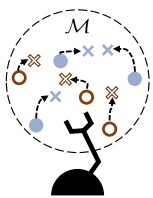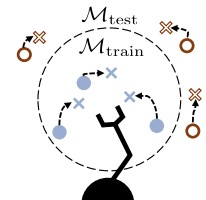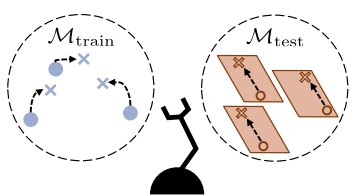

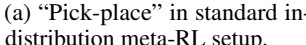

(a) "Pick-place" in standard in-distribution meta-RL setup.

(b) "Pick-place" in out-of-distribution meta-RL setup.

(c) Non-parametric task variation between "Pick-place" and "Sweep-into."

Figure 1: **Problem Setup.** Visualizing different meta-RL scenarios with Meta-World tasks [69, 71]. The circles and crosses represent the object and goal positions, respectively. Solid blue objects indicate training tasks, while empty brown objects indicate test tasks.

Moreover, even if an agent successfully models the inductive bias parametrically, it is improbable that the same parameterization will be reusable for qualitatively distinct test tasks.

In addressing the challenges of non-parametric task variability in meta-RL, our primary strategy involves *decomposing each non-parametric task into a set of shared elementary subtasks*. We then parameterize each task based on the types of subtasks that constitute it. Despite the non-parametric task variability, tasks may share elementary subtasks. For instance in Figure 1c, a "Pick-place" task can be decomposed into subtasks: "grip object" and "place object," while a "Sweep-into" task can be decomposed into subtasks: "grip object" and "push object." By employing the shared subtask parameterization, the policy can capitalize on the captured commonalities between non-parametric tasks to enhance training efficiency and generalization capabilities.

However, our approach of task parameterization based on a set of subtasks faces two primary challenges: the lack of prior information about **(1)** the set of elementary subtasks and **(2)** the decomposition of each task. To address these issues, we employ meta-learning for the subtask decomposition (SD) process using a Gaussian mixture variational autoencoder (GMVAE) [29, 8, 59]. Our GMVAE encodes the trajectory up to the current timestep into latent categorical and Gaussian contexts, which are trained to reconstruct the task's reward and transition dynamics [74]. We discover that the meta-learned latent categorical context effectively represents the subtask compositions of tasks under non-parametric variations. Consequently, the policy, using the learned subtask composition, can readily generalize to new tasks comprising previously encountered subtasks. To further enhance generalization to unseen compositions of familiar subtasks, we propose a virtual training (VT) process [35, 1] specifically designed for non-parametric task variability. We train the policy on imaginary tasks generated by the learned dynamics decoder, conditioned on hypothetical subtask compositions.

We evaluate our method on the Meta-World ML-10 and ML-45 benchmarks [71], widely used meta-RL benchmarks comprising diverse non-parametric robotic manipulation tasks. We empirically demonstrate that our method successfully meta-learns the shareable subtask decomposition. With the help of the subtask decomposition and virtual training, our method, without any offline demonstration or test-time gradient updates, achieves test success rates of 33.4% on ML-10 and 31.2% on ML-45, which improves the previous state-of-the-art by approximately 1.7 times and 1.3 times, respectively.

## 2 Background

### 2.1 Meta-Reinforcement Learning

A Markov decision process (MDP), $M = (\mathcal{S}, \mathcal{A}, R, T, T_0, \gamma, H)$, is defined by a tuple comprising a set of states $\mathcal{S}$, a set of actions $\mathcal{A}$, a reward function $R(r_{t+1}|s_t, a_t, s_{t+1})$, a transition function $T(s_{t+1}|s_t, a_t)$, an initial state distribution $T_0(s_0)$, a discount factor $\gamma$, and a horizon $H$.

The goal of meta-RL is to learn to adapt to a distribution of MDPs with varying reward and transition dynamics. At the start of each meta-training iteration, an MDP is sampled from the distribution $p(\mathcal{M}_{\text{train}})$ over a set of MDPs $\mathcal{M}_{\text{train}}$. Each MDP $M_k = (\mathcal{S}, \mathcal{A}, R_k, T_k, T_{0,k}, \gamma, H)$ is defined with a unique reward function $R_k$, transition function $T_k$, and initial state distribution $T_{0,k}$. Unlike in a multi-task setup, the agent in the meta-RL setup does not have access to the task index $k$ that determines the MDP dynamics. The training objective is to optimize the policy $\pi_\psi$ with

parameters $\psi$ to maximize the expected return across all MDPs: $\max_\psi \mathbb{E}_{M_k \sim p(\mathcal{M}_{\text{train}})} [\mathcal{J}_{\text{pol}}(\psi)]$, where $\mathcal{J}_{\text{pol}}(\psi) = \mathbb{E}_{T_{0,k}, T_k, \pi_\psi} \left[ \sum_{t=0}^{H-1} \gamma^t R_k(r_{t+1}|s_t, a_t, s_{t+1}) \right]$. During meta-testing, standard in-distribution meta-RL methods are evaluated on tasks sampled from the same distribution $p(\mathcal{M}_{\text{test}})$ as the training tasks, i.e., $\mathcal{M}_{\text{train}} = \mathcal{M}_{\text{test}} = \mathcal{M}$ (Figure 1a). In contrast, OOD meta-RL methods assume strictly disjoint training and test task sets, i.e., $\mathcal{M} = \mathcal{M}_{\text{train}} \cup \mathcal{M}_{\text{test}}$ and $\mathcal{M}_{\text{train}} \cap \mathcal{M}_{\text{test}} = \emptyset$ (Figure 1b).

## 2.2 Bayes-adaptive Meta-Reinforcement Learning

Since the true task index $k$ is not provided to the agent in the meta-RL problem setup, it is important to balance exploration and exploitation while learning about the initially unknown MDP. A Bayes-adaptive agent [38, 10, 16] achieves this balance by updating its belief $b_t(R, T)$ about the MDP based on its experience $\tau_{:t} = \{s_0, a_0, r_1, s_1, a_1, r_2, \ldots, s_{t-1}, a_{t-1}, r_t, s_t\}$. The agent's belief over the MDP dynamics at time $t$ can be represented as a posterior given the trajectory, i.e., $b_t(R, T) = p(R, T|\tau_{:t})$. By augmenting the state with the belief to form a hyper-state space $\mathcal{S}^+ = \mathcal{S} \times \mathcal{B}$, where $\mathcal{B}$ is the set of belief, a Bayes-adaptive MDP (BAMDP) can be constructed. The objective of a BAMDP is to maximize the expected return within a meta-episode while learning, where a meta-episode consists of $n_{\text{roll}}$ rollout episodes (i.e., $H^+ = n_{\text{roll}} \times H$ steps) of the same MDP:

$$\mathcal{J}_{\text{pol}}^+(\psi) = \mathbb{E}_{b_0, T^+, \pi_\psi} \left[ \sum_{t=0}^{H^+-1} \gamma^t R^+(r_{t+1}|s_t^+, a_t, s_{t+1}^+) \right], \tag{1}$$

where $T^+(s_{t+1}^+|s_t^+, a_t, r_t) = \mathbb{E}_{b_t} [T(s_{t+1}|, s_t, a_t)] \, \mathbb{I}(b_{t+1} = p(R, T|\tau_{:t+1}))$ is the transition dynamics and $R^+(r_{t+1}|s_t^+, a_t, s_{t+1}^+) = \mathbb{E}_{b_{t+1}} [R(r_{t+1}|s_t, a_t, s_{t+1})]$ is the reward dynamics of the BAMDP. The posterior belief update in the indicator function $\mathbb{I}(\cdot)$ is intractable for all but simple environments.

VariBAD [74] solves the inference and posterior update of the belief by combining meta-RL and approximate variational inference [28]. At each timestep $t$, a recurrent encoder $q_{\phi_h}$ encodes the experience $\tau_{:t}$ into a hidden embedding $h_t = q_{\phi_h}(\tau_{:t})$. The approximate posterior belief over the dynamics can be represented as the parameters of a multivariate Gaussian distribution: $b_t = (\mu_{\phi_z}(h_t), \sigma_{\phi_z}(h_t))$, where $\mu_{\phi_z}(\cdot)$ and $\sigma_{\phi_z}(\cdot)$ are neural networks. The latent context $z_t \sim \mathcal{N}(\mu_{\phi_z}(h_t), \sigma_{\phi_z}^2(h_t))$ is used to estimate the MDP dynamics: $p_{\theta_R}(r_{j+1}|s_j, a_j, s_{j+1}, z_t)$ and $p_{\theta_T}(s_{j+1}|s_j, a_j, z_t)$ for $j = 0, \ldots, H^+ - 1$, *including the past and future*. Then the problem of computing the posterior over the dynamics $p(R, T|\tau_{:t})$ reduces to inferring the posterior $q_\phi(z_t|\tau_{:t})$, where $\phi = \{\phi_h, \phi_z\}$. A separate policy network $\pi_\psi(a_t|s_t, b_t)$ is trained to optimize the BAMDP objective in Eq. (1).

## 2.3 Non-parametric Task Variability

The term "non-parametric" in the context of task variability is introduced in the Meta-World paper [69]. It is used to distinguish the task variability in Meta-World manipulation tasks from more simplistic parametric variations exhibited in standard MuJoCo tasks, such as variations in target goal positions, directions, and velocities. However, the term "non-parametric" might lead to misunderstanding. Because it could suggest that there are no parameters available for task parameterization, even though we have effectively parameterized them in our approach. Our major breakthrough is in enabling this parameterization in terms of subtask compositions, which was difficult using earlier methods that relied on simple latent parametrization. Hence, while we retain the "non-parametric" terminology, we guide the readers to view the scope of our work through the lens of modularity or composition-based generalization [27].

# 3 Method

In this section, we introduce our novel meta-RL method, named Subtask Decomposition and Virtual Training (SDVT), to handle non-parametric task variability. Our approach is based on decomposing each task into a set of elementary subtasks and parameterizing each task based on the composition of these subtasks. To achieve this, we use a Gaussian mixture variational autoencoder (GMVAE) to meta-learn the subtask decomposition process. In addition, we introduce a virtual training process that improves generalization to tasks with unseen compositions of subtasks.

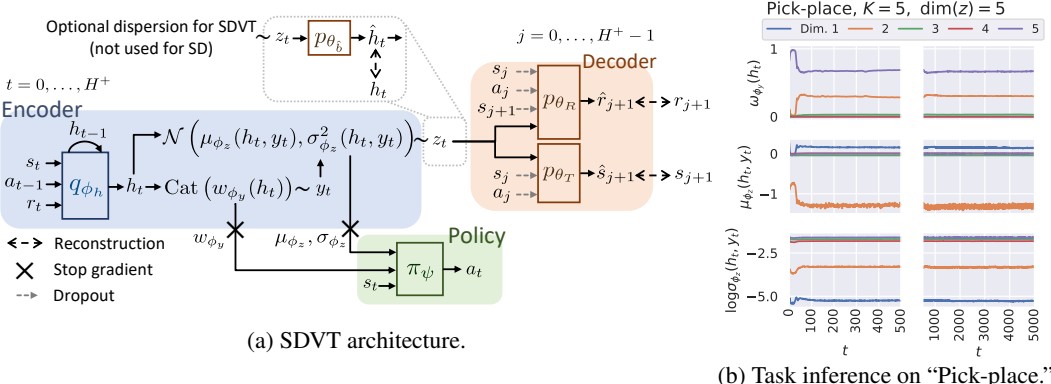

(a) SDVT architecture.

(b) Task inference on "Pick-place."

Figure 2: **SDVT architecture. (a)** Our proposed architecture incorporates three main components: the encoder, decoder, and policy. An online trajectory is encoded into categorical ($y$) and Gaussian ($z$) latent contexts. These contexts, which are trained to reconstruct the forward dynamics, are utilized by the policy network. This structure is also applied to the virtual training, as illustrated in Figure 3a, with an optional dispersion layer integrated. **(b)** An example of the learned task inference process within a meta-episode ($H^+ = 5000$ steps) on "Pick-place" is shown. We report the values for each dimension of contexts: $\omega_{\phi_y}(h_t)$, $\mu_{\phi_z}(h_t, y_t)$, and $\log \sigma_{\phi_z}(h_t, y_t)$.

## 3.1 Subtask Decomposition (SD) with a Gaussian Mixture Variational Autoencoder

Our goal is to meta-learn a set of elementary subtasks and to meta-learn the decomposition of each task into a composition of these subtasks. The core of our method focuses on meta-learning the approximate subtask composition $y_t \in \Delta^K$ sampled from a $K$-class categorical distribution, where $\Delta^K$ denotes the $K$-dimensional probability simplex. For example with $K = 3$, we want the subtask composition $y_t$ to be learned somewhat like $(0.5, 0.5, 0.0)$ for "Pick-place" and $(0.0, 0.5, 0.5)$ for "Sweep-into," where each dimension of $y_t$ represents the weight corresponding to subtasks in the order of "place object," "grip object," and "push object." To capture such subtask information, we use a Gaussian mixture variational autoencoder (GMVAE) to represent the latent space of non-parametric tasks as a Gaussian mixture distribution. Each task is represented as a $K$-dimensional mixture proportion $y_t$, resulting in each class representing a unique subtask shared across different tasks. Thus, the distribution of $K$ subtasks is modeled using a categorical distribution with $K$ classes, where each class is associated with a Gaussian distribution. The learned subtask composition $y_t$ represents the agent's belief at time $t$ about the current task's decomposition into subtasks. It is crucial to note that we don't necessarily want $y_t$ to be a one-hot embedding representing the subtask that the agent is solving at time $t$. We want $y_t$ to represent a mixing proportion [56] over all possible subtasks that the agent believes to be relevant to the current task, including past and future as in Figures 2b and 4. This distinction is vital to the model's effectiveness in solving a range of tasks, as it allows for flexibility in subtask identification and generalization across different tasks.

**Architecture**  Our full model in Figure 2a, which is based on the VAE of VariBAD [74], consists of three main components: the encoder, decoder, and policy networks parameterized by $\phi = \{\phi_h, \phi_y, \phi_z\}$, $\theta = \{\theta_z, \theta_R, \theta_T\}$, and $\psi$, respectively.

**(1)** The encoder is defined as $q_\phi(y_t, z_t|h_t) = q_{\phi_y}(y_t|h_t) q_{\phi_z}(z_t|h_t, y_t)$. A recurrent network encodes the past trajectory $\tau_{:t}$ into a hidden embedding $h_t = q_{\phi_h}(\tau_{:t})$. First, the categorical encoder $q_{\phi_y}(y_t|h_t) : \text{Cat}(\omega_{\phi_y}(h_t))$ samples $y_t$, where $\omega_{\phi_y}(h_t) \in \Delta^K$. We use the Gumbel-Softmax trick [23] with a high temperature ($\tau = 1$) when sampling $y_t$ to form a soft label. Then the multivariate Gaussian encoder $q_{\phi_z}(z_t|h_t, y_t) : \mathcal{N}(\mu_{\phi_z}(h_t, y_t), \sigma^2_{\phi_z}(h_t, y_t))$ samples a continuous latent context $z_t$, which contains the parametric information of the subtasks, in addition to the categorical subtask information of $y_t$.

**(2)** The decoder is defined as $p_\theta(\tau_{:H^+}, y_t, z_t) = p(y_t) p_{\theta_z}(z_t|y_t) p_{\theta_R, \theta_T}(\tau_{:H^+}|z_t)$. It reconstructs the reward and transition dynamics for all transitions in the meta-episode $\tau_{:H^+}$, using the latent context $z_t$ as in Eq. (4). We assume a uniform prior of subtask composition $p(y_t) : \text{Uniform}(1/K)$ and a Gaussian regularization $p_{\theta_z}(z_t|y_t) : \mathcal{N}(\mu_{\theta_z}(y_t), \sigma^2_{\theta_z}(y_t))$. This encoder-decoder architecture

allows both the approximate posterior $q_\phi(y_t, z_t|h_t)$ and the prior $p(z_t)$ to follow Gaussian mixture distributions.

**(3)** The policy network, $\pi_\psi(a_t|s_t, b_t)$, is trained separately conditioned on the belief $b_t = (\mu_{\phi_z}(h_t, y_t), \sigma_{\phi_z}(h_t, y_t))$ that are parameters of the Gaussian context. In practice, we also provide the parameters of the categorical encoder $\omega_{\phi_y}(h_t)$ to the policy, which we find to improve the performance. The parameters of the distributions ($\omega_{\phi_y}, \mu_{\phi_z}, \sigma_{\phi_z}, \mu_{\theta_z}$, and $\sigma_{\theta_z}$) are modeled as outputs of multilayer perceptrons (MLPs) as in Appendix C.4.

**Objective** We optimize the GMVAE to maximize the evidence lower bound (ELBO) for all time steps $t = 0, \ldots, H^+$ over the trajectory distribution $d(M_k, \tau_{:H^+})$ induced by the policy in MDP $M_k$:

$$\text{ELBO}_t(\phi, \theta) = \mathbb{E}_{d(M_k, \tau_{:H^+})} \left[ \mathbb{E}_{q_\phi(y_t, z_t|h_t)} \mathcal{J}_{\text{GMVAE}} \right], \tag{2}$$

$$\mathcal{J}_{\text{GMVAE}} = \alpha_R \mathcal{J}_{\text{R-rec}} + \alpha_T \mathcal{J}_{\text{T-rec}} + \alpha_g \mathcal{J}_{\text{reg}} + \alpha_c \mathcal{J}_{\text{cat}}. \tag{3}$$

In addition to the reconstruction objectives $\mathcal{J}_{\text{R-rec}}$ and $\mathcal{J}_{\text{T-rec}}$, we have additional regularization $\mathcal{J}_{\text{reg}}$ and categorical $\mathcal{J}_{\text{cat}}$ objectives:

$$\mathcal{J}_{\text{R-rec}} = \sum_{j=0}^{H^+-1} \log p_{\theta_R}(r_{j+1}|s_j, a_j, s_{j+1}, z_t), \quad \mathcal{J}_{\text{T-rec}} = \sum_{j=0}^{H^+-1} \log p_{\theta_T}(s_{j+1}|s_j, a_j, z_t), \tag{4}$$

$$\mathcal{J}_{\text{reg}} = \log \frac{p_{\theta_z}(z_t|y_t)}{q_{\phi_z}(z_t|h_t, y_t)}, \quad \mathcal{J}_{\text{cat}} = \log \frac{p(y_t)}{q_{\phi_y}(y_t|h_t)}. \tag{5}$$

The regularization objective $\mathcal{J}_{\text{reg}}$ minimizes the KL divergence between the learned posterior Gaussian distribution $q_{\phi_z}(z_t|h_t, y_t)$ and learned Gaussian priors $p_{\theta_z}(z_t|y_t)$. Unlike the standard VAE that assumes a standard normal prior, we learn $K$ distinct Gaussian priors conditioned on $y_t$. The categorical objective $\mathcal{J}_{\text{cat}}$ maximizes the conditional entropy of $y_t$ given $h_t$ and prevents collapse. Refer to Appendix B for the derivation of the ELBO objective. The reconstruction objectives in Eq. (4) are computed for all timesteps from the first to the terminal step of the meta-episode. Therefore, the subtask composition $y_t$ at time $t$ is not necessarily a one-hot label of the belief about the current subtask at time $t$, but a mixture label of the belief on all subtasks that compose the current task in the past and future within the meta-episode. Under the BAMDP setup, the agent learns to reduce the uncertainty of the decomposition as quickly as possible, supporting the policy with the converged parameterization. Combining the policy objective in Eq. (1) and the sum of the ELBO objectives in Eq. (2) for all timesteps in a meta-episode, the overall objective over training tasks is to maximize:

$$\mathcal{J}(\phi, \theta, \psi) = \mathbb{E}_{M_k \sim p(\mathcal{M}_{\text{train}})} \left[ \mathcal{J}_{\text{pol}}^+(\psi) + \sum_{t=0}^{H^+} \text{ELBO}_t(\phi, \theta) \right]. \tag{6}$$

**Occupancy regularization** The dimension of $y_t$ or the number of underlying subtasks $K$ is a crucial hyperparameter that should be determined based on the number of training tasks $N_{\text{train}}$ and their similarities. However, in many cases, prior information about the optimal value of $K$, denoted as $K^*$, may not be available. One way to expand the scope of our method for unknown $K^*$ is to meta-learn the number of effective subtasks as well. First, we assume $K^* < N_{\text{train}}$, otherwise each task will be classified into a separate subtask with one-hot label, preventing learning shareable subtasks. We start with a sufficiently large $K = N_{\text{train}}$ and regularize the ELBO objective to progressively reduce the number of effective subtasks (non-zero components) occupied in $y_t$ with the following occupancy regularization that penalizes the usage of larger indices in the subtask composition:

$$\mathcal{J}_{\text{occ}} = -\log K \left( e^{-K+1}, e^{-K+2}, \ldots, e^{-1}, e^0 \right) \cdot y_t. \tag{7}$$

We calculate the dot product of exponential weights and the subtask composition $y_t$ to penalize the occupancy of higher dimensions of $y_t$. We scale the dot product by $\log K$ to match the scale to the upper bound of $\mathcal{J}_{\text{cat}}$. We add $\mathcal{J}_{\text{occ}}$ multiplied by a coefficient $\alpha_o$ to the GMVAE objective in Eq. (3). Consequently, the agent prioritizes using lower indices in the decomposition to represent frequent subtasks and sparingly uses higher indices for rare subtasks as in Figure 4b and in Appendix E.1.

**Decoder dropout** As the GMVAE is optimized using the trajectories induced by the policy, the decoder can easily overfit the frequent states and actions of training tasks [35]. This can lead to low predicted rewards for unexperienced states and actions, regardless of the latent context $z_t$. When

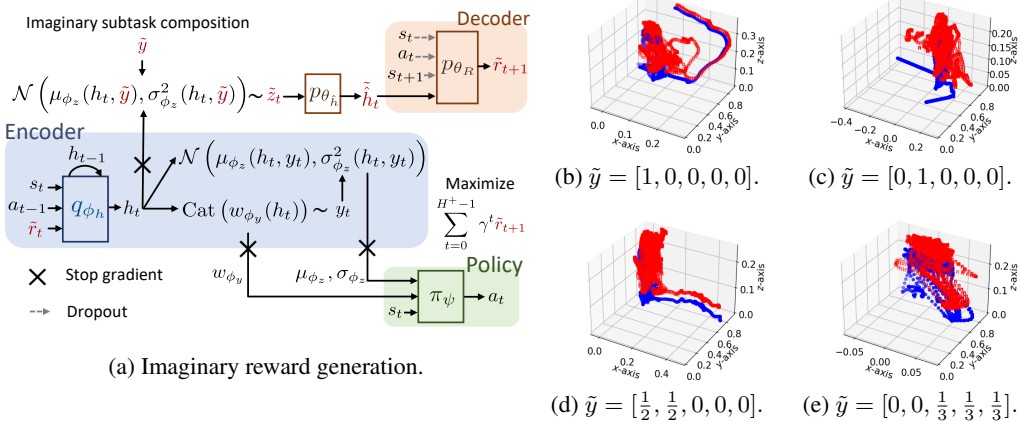

(a) Imaginary reward generation.

(b) $\tilde{y} = [1, 0, 0, 0, 0]$.

(c) $\tilde{y} = [0, 1, 0, 0, 0]$.

(d) $\tilde{y} = [\frac{1}{2}, \frac{1}{2}, 0, 0, 0]$.

(e) $\tilde{y} = [0, 0, \frac{1}{3}, \frac{1}{3}, \frac{1}{3}]$.

Figure 3: **Virtual training. (a)** Generation of imaginary rewards with the decoder conditioned on a fixed imaginary subtask composition $\tilde{y}$. **(b)–(e)** Examples depicting the diversity of generated imaginary tasks, where $K = 5$ and all states are from the Meta-World "Reach." Red rods and blue circles represent the trajectories of the gripper and object, respectively. These trajectories, while generated by the same policy, differ across various imaginary subtask compositions $\tilde{y}$.

such overfitting happens, the latent context loses its task-informative value, leading to the potential underperformance of policy based on this context. Following the approach of LDM [35], we address this by applying dropout (DO) to the state and action embeddings of the decoder while leaving latent context $z_t$ untouched: $p_{\theta_R}(r_{j+1}|\text{DO}[s_j, a_j, s_{j+1}], z_t)$ and $p_{\theta_T}(s_{j+1}|\text{DO}[s_j, a_j], z_t)$. This dropout application is crucial for the virtual training discussed in the following section.

## 3.2 Virtual Training (VT) on Generated Tasks with Imaginary Subtask Compositions

The overall objective in Eq. (6) is optimized over the trajectories of training tasks. To enhance generalization to test tasks with unseen subtask compositions, we generate virtual training tasks using the imaginary dynamics produced by the GMVAE decoder. This process resembles those in [35, 1], but their generated tasks are limited to parametric variations, e.g., generating tasks with unseen goal positions. Such methods can prepare for test tasks with unseen parametric changes but struggle to prepare for qualitatively new tasks with unseen compositions of subtasks. Our GMVAE model enables us to extend the process to the non-parametric setup by conditioning the decoder on an imaginary subtask composition $\tilde{y}$. At the beginning of each meta-episode, we randomly determine with probability $p_v$ whether to convert it into a virtual meta-episode. By training the policy on imaginary tasks, it can better prepare for test tasks with unseen subtask compositions in advance. We use the tilde accent to denote imaginary components, and the hat accent to denote estimates.

**Latent context dispersion** Inspired by the work of Ajay et al. [1], we utilize the dispersion structure for our GMVAE to support extrapolated generalization of the latent context $z_t$. Instead of directly using $z_t$ as the decoder input, we insert an additional MLP $p_{\theta_{\hat{h}}}$ before the decoder to expand the dimension of the context (the dotted box in Figure 2a). The MLP output $\hat{h}_t = p_{\theta_{\hat{h}}}(z_t)$ is trained to reconstruct the embedding $h_t$ by appending $\alpha_d \mathcal{J}_{\text{dis}} = -\alpha_d \|h_t - \hat{h}_t\|_2^2$ to the total GMVAE objective in Eq. (3). We then use the dispersed context $\hat{h}_t$ instead of $z_t$ to reconstruct the reward and transition. This trick is effective in generating imaginary tasks featuring extrapolated dynamics, albeit at the cost of increased training complexity.

**Imaginary reward generation** Figure 3a presents the imaginary reward generation process. The goal is to create imaginary tasks based on the distribution of training subtasks but with unseen compositions of subtasks. At the beginning of a virtual meta-episode, we randomly sample an imaginary subtask composition $\tilde{y} \sim \text{Dirichlet}(\bar{y})$ fixed for that virtual meta-episode, where the concentration parameter $\bar{y} \in \Delta^K$ is the empirical running mean of all $y_t$ over training. By replacing the real $y_t$ with the imaginary subtask composition $\tilde{y}$, we sample an imaginary context $\tilde{z}_t \sim \mathcal{N}(\mu_{\phi_z}(h_t, \tilde{y}), \sigma^2_{\phi_z}(h_t, \tilde{y}))$, imaginary dispersed context $\tilde{\hat{h}}_t = p_{\theta_{\hat{h}}}(\tilde{z}_t)$, and finally the imagi-

nary reward $\tilde{r}_{t+1} \sim p_{\theta_R}(r_{t+1}|\text{DO}\,[s_t, a_t, s_{t+1}], \tilde{\tilde{h}}_t)$ accordingly using our GMVAE. We replace the reward for the next timestep, $r_{t+1}$, with $\tilde{r}_{t+1}$, while the states remain to be from the real training task. The imaginary reward is used for the encoder input at the next time step and for the policy, where the policy is trained to maximize the sum of generated rewards, $\sum_{t=0}^{H^+-1} \gamma^t \tilde{r}_{t+1}$. However, the GMVAE is not trained to reconstruct the imaginary dynamics.

## 3.3 Summary of the Combined Methods: SDVT-LW and SDVT

By combining the subtask decomposition (SD) and virtual training (VT) processes, we propose two methods: SDVT-LW and SDVT. Foremost, our primary contribution is the proposal of SDVT-LW, which is the lightweight (-LW) version of our method that assumes the prior knowledge of the optimal number of subtasks $K$, therefore not employing the occupancy regularization.

Furthermore, we propose SDVT with the occupancy regularization strategy. This generalizes SDVT-LW to adapt to more difficult conditions where there is a lack of prior knowledge of the optimal number of subtasks $K^*$. We initialize the number of subtasks equal to the number of training tasks and employ occupancy regularization to downscale higher dimensions, navigating to discover the most efficient number of subtasks even without prior knowledge.

For the purposes of virtual training, we adopt two methodologies that have found application in previous studies: dropout [35] and dispersion via structured VAE [1]. Their use in our work remains unchanged from their original applications. The efficacy of these components, within the context of virtual training, is demonstrated via ablations in Appendix E. We summarize the entire meta-training process with a pseudocode in Appendix A.

## 4 Related Work

**Classical meta-RL** Classical meta-RL methods assume that both training and test tasks are sampled from the same distribution. These methods are divided into two main categories: gradient-based methods and context-based methods. Gradient-based methods [13, 54, 48, 73, 53] learn a common initialization of a lightweight model for all tasks, allowing the agent to achieve high performance on unseen target tasks with a few steps of gradient updates. However, these methods lack online adaptation capability within a meta-episode because they require many pre-update rollouts before adaptation. Context-based methods [9, 18, 46, 32, 34, 43, 74, 40] use a recurrent or memory-augmented network to encode the collected experience into a latent context. In general, the context is trained to optimize auxiliary objectives such as reward dynamics, transition dynamics, and value function. These methods can adapt to target tasks through online, in-context learning without requiring gradient updates. However, they are vulnerable to test-time distribution shifts since the encoded context and the policy given the context are hardly generalized to out-of-distribution tasks.

**Out-of-distribution meta-RL** A group of recent studies focuses on training a generalizable agent that is robust to test-time distribution shifts. A group of works generates imaginary observations using image augmentation techniques [18, 21, 31, 44, 33, 61, 66]. Most of these methods depend on predefined heuristic augmentations, without utilizing the training task dynamics. On the other hand, some works explicitly address varying environment dynamics. For example, MIER [7] reuses the trajectories collected during training by relabeling according to the test dynamics. Our work is related to AdMRL [36], LDM [35], and DiAMetR [1], which generate imaginary tasks with unseen dynamics using learned models. However, these works focus on generating parametric variants of training tasks, while we focus on generalizing across non-parametric task variants.

**Skill-based meta-RL** Our task parameterization based on the subtask decomposition is related to the recently spotlighted skill-based meta-RL methods [11, 51, 50, 55], which aim to achieve fast generalization on unseen tasks by decomposing and extracting reusable skills from training tasks' trajectories. These works often require refined offline demonstrations to learn skills using behavioral cloning objectives, where the skills distinguish a sequence of actions given a sequence of states. For example, SimPL [42] extracts skills from offline demonstrations before meta-training, and during a meta-test, it only adapts the high-level skill policy, with the low-level policy frozen. HeLMS [47] learns a 3-level skill hierarchy by decomposing offline demonstrations of a single task. Online learning of the skills is often unstable because the set of skills should develop along with the online improvement of the policy. The subtask decomposition of our method is conceptually different from

the skill decomposition [68]. Even when the policy is not stationary during online updates, the underlying reward and transition dynamics, which our model has to estimate, do not change.

# 5  Experiments

## 5.1  Experimental Setup

**Meta-World benchmark**  The Meta-World V2 benchmark [71] stands as the most prominent, if not the only, established benchmark for assessing meta-RL algorithms featuring non-parametric task variability. This benchmark comprises 50 qualitatively distinct robotic manipulation tasks, with each task containing 50 parametric variants that incorporate randomized goals and initial object positions. Specifically, the Meta-World Meta-Learning 10 (ML-10) benchmark, consists of $N_{\text{train}} = 10$ training tasks and $N_{\text{test}} = 5$ held-out test tasks. We denote each task by an index, where the training tasks are numbered from 1 to 10 and the test tasks from 11 to 15. Likewise, the ML-45 benchmark consists of $N_{\text{train}} = 45$ training tasks and $N_{\text{test}} = 5$ test tasks. Refer to the tables in Appendix D.1 for the set and indices of tasks. The agent must maximize its return from experience while exploring to identify the initially unknown task dynamics within a meta-episode of $H^+ = 5000$ steps that consists of $n_{\text{roll}} = 10$ rollout episodes of horizon $H = 500$ steps each.

**SDVT variants and baselines setup**  We evaluate our methods SDVT and SD (only subtask decomposition without virtual training and dispersion). Without the prior knowledge of $K^*$, we set $K = N_{\text{train}}$ and apply the occupancy regularization ($\alpha_o = 1.0$) with $\alpha_c = 1.0$. We also evaluate a lightweight (-LW) version of ours with smaller $K = 5$ with $\alpha_c = 0.5$ and without the occupancy regularization ($\alpha_o = 0.0$). To ensure a fair comparison and to exclude the gains from orthogonal contributions, we compare SDVT with state-of-the-art meta-RL methods that do not require any refined offline demonstrations, ensembles, or extensive test-time training: RL$^2$ [9], MAML [13], PEARL [46], and VariBAD [74]. We also compare with a parametric OOD meta-RL method, LDM [35], to evaluate the efficacy of our virtual training over subtasks. All methods do not perform gradient updates during the test except for MAML. Appendix C presents more implementation details. Briefly, SDVT without a Gaussian mixture reduces to LDM, LDM without virtual training reduces to VariBAD, and VariBAD without a VAE decoder reduces to RL$^2$. Our implementation is available at `https://github.com/suyoung-lee/SDVT`.

**Evaluation metric**  We follow the standard success criterion of Meta-World as follows. A timestep is considered successful if the distance between the task-relevant object and the goal is less than a predefined threshold. A rollout episode is considered successful if the agent ever succeeds at any timestep during the rollout episode. The success of a meta-episode is defined as the success of the last (10th) rollout episode. In Table 1, we report the success rates of 750 (15 tasks $\times$ 50 parametric variants) meta-episodes at 250M steps for ML-10 and 2500 (50 tasks $\times$ 50 parametric variants) meta-episodes at 400M steps for ML-45.[2] Likewise, we report the returns of the last rollout episodes.

## 5.2  Results

Table 1 illustrates that no method attains a 100% success rate even on training tasks, emphasizing the challenge posed by non-parametric task variability. The training success rates on ML-45 are consistently lower than those on ML-10, reflecting the inherent difficulty in adapting to a broader range of tasks, such as conflicting gradients [70]. Notably, SD surpasses all other baselines in training success rate. In particular, outperforming VariBAD underscores the limitations of employing a single Gaussian distribution to model the latent task space in cases of non-parametric variability.

On the test tasks, SDVT and SDVT-LW substantially outperform all baselines, even outperforming LDM, which surpasses VariBAD with parametric virtual training. Our gain is attributed to our virtual training process, which is specifically designed for test tasks involving non-parametric variability. Notably, SDVT and SD outperform their LW counterparts in training success rates, primarily due to its fine-grained subtask decomposition which provides a more precise representation of each task. For example, a "grip object" subtask may be split and represented as a combination of two distinct subtasks "move gripper" and "tighten gripper" with a larger $K$. In contrast, SDVT-LW scores higher

---

[2]Our aggregation method diverges from the atypical method employed in the Meta-World paper [71], which presents the main results as the average of the *maximum* success rate of each task (details in Appendix C.5)

Table 1: **Meta-World V2 success rates and returns.** We report the final success rates (%) and returns of our methods and baselines averaged across training tasks and test tasks of the ML-10 and ML-45 benchmarks. All results are reported as the mean success rate $\pm$ 95% confidence interval of 8 seeds. Individual scores of all tasks are reported in Appendix D.1.

| | Success Rate | | | | Return | | | |
| | ML-10 | | ML-45 | | ML-10 | | ML-45 | |
| Methods | Train | Test | Train | Test | Train | Test | Train | Test |
|---|---|---|---|---|---|---|---|---|
| SDVT | **77.2±3.0** | 32.8±3.9 | 55.6±4.2 | 28.1±3.2 | **3656±62** | 1225±160 | 2379±214 | 839±74 |
| SDVT-LW | 62.1±4.1 | **33.4±5.0** | 50.4±4.1 | **31.2±1.2** | 3454±137 | **1527±214** | 2294±202 | **894±27** |
| SD | 77.0±5.9 | 30.8±7.7 | **61.0±1.7** | 23.0±5.1 | 3630±241 | 1112±190 | **2672±79** | 786±69 |
| SD-LW | 75.5±5.5 | 26.2±8.7 | 56.7±1.5 | 25.4±2.9 | 3525±297 | 1043±234 | 2578±64 | 793±49 |
| RL$^2$ | 67.4±4.4 | 15.1±2.7 | 58.0±0.4 | 11.8±3.2 | 1159±83 | 715±33 | 1411±22 | 663±100 |
| MAML | 42.2±4.5 | 3.9±3.7 | 32.0±1.4 | 19.8±6.3 | 1822±136 | 439±78 | 1388±104 | 658±96 |
| PEARL | 23.2±1.9 | 0.8±0.5 | 10.3±2.4 | 6.7±3.3 | 1081±77 | 340±54 | 597±121 | 506±122 |
| VariBAD | 58.2±8.9 | 14.1±6.1 | 57.0±1.2 | 22.1±3.5 | 3055±466 | 919±143 | 2492±47 | 762±40 |
| LDM | 56.7±12.3 | 19.8±6.0 | 54.1±0.9 | 24.8±2.9 | 2963±626 | 1166±264 | 2515±67 | 768±63 |

test success rates than SDVT, presumably due to its coarser decomposition that allows imaginary compositions to encompass a broader range of unseen test tasks.

Please refer to the tables in Appendix D.1 for individual task results. Our methods achieve the highest success rates across all test tasks on the ML-10 benchmark. However, all methods encounter challenges in solving "Shelf-place," which includes an unseen shelf not present in the observation. As such, this task cannot be decomposed into previously seen subtasks but rather into unseen ones, making it difficult to prepare through virtual training. These tasks, composed of unseen subtasks, pose a considerable challenge for zero-shot adaptation as with SDVT. Addressing these tasks may require a substantial increase in rollouts and updates during tests, an area of potential future work. Detailed ablation results are reported in Appendix E. For additional results such as rendered videos, refer to our webpage `https://sites.google.com/view/sdvt-neurips`.

## 5.3 Analysis

In this subsection, we explore whether the performance enhancements attributed to our methods are indeed the result of effectively decomposed subtasks and a diverse range of imaginary tasks.

**Learned Subtask Compositions** From Figure 2b, we observe a rapid context convergence within the first rollout episode for a sufficiently meta-learned task. To validate our motivation, we visualize the converged subtask compositions in Figure 4. We find that such converged subtask compositions are shared by qualitatively similar tasks. For example, in Figure 4a, tasks (3) "Pick-place," (7) "Peg-insert-side," and (10) "Basketball," which require placing an object at a goal position, share subtask indices 1 and 3. Additionally, simple tasks that require moving the gripper in a straight line: (1) "Reach," (5) "Drawer-close," and (8) "Window-open" are classified into the same subtask 2. These observations suggest that our model can effectively meta-learn the decomposition process of tasks into shareable subtasks. Furthermore, the test tasks may not necessarily have the same subtask compositions as the training tasks. For instance, (3) "Pick-place" and (14) "Sweep-into" share subtask 3, but not subtasks 1 or 5, revealing the potential for virtual training to be effective.

**Occupancy Regularization** Figure 4 indicates that it is important to set the subtask class dimension $K$ and the categorical coefficient $\alpha_c$ appropriately according to the number of training tasks and the correlations among them. Otherwise, the decomposition may suffer from a collapse or degeneration. With the occupancy regularization, we can avoid the extensive search for the optimal subtask dimension $K^*$ and corresponding $\alpha_c$. In Figure 4b, we find that our occupancy regularization successfully limits the use of unnecessary subtasks with higher indices as intended.

**Generated imaginary tasks** To demonstrate the dynamics of the imaginary tasks, we show the trajectory of the gripper and object over a meta-episode (5000 steps) on generated imaginary tasks for different imaginary subtask compositions $\tilde{y}$ in Figure 3. When we set $\tilde{y}$ as a one-hot vector, the policy

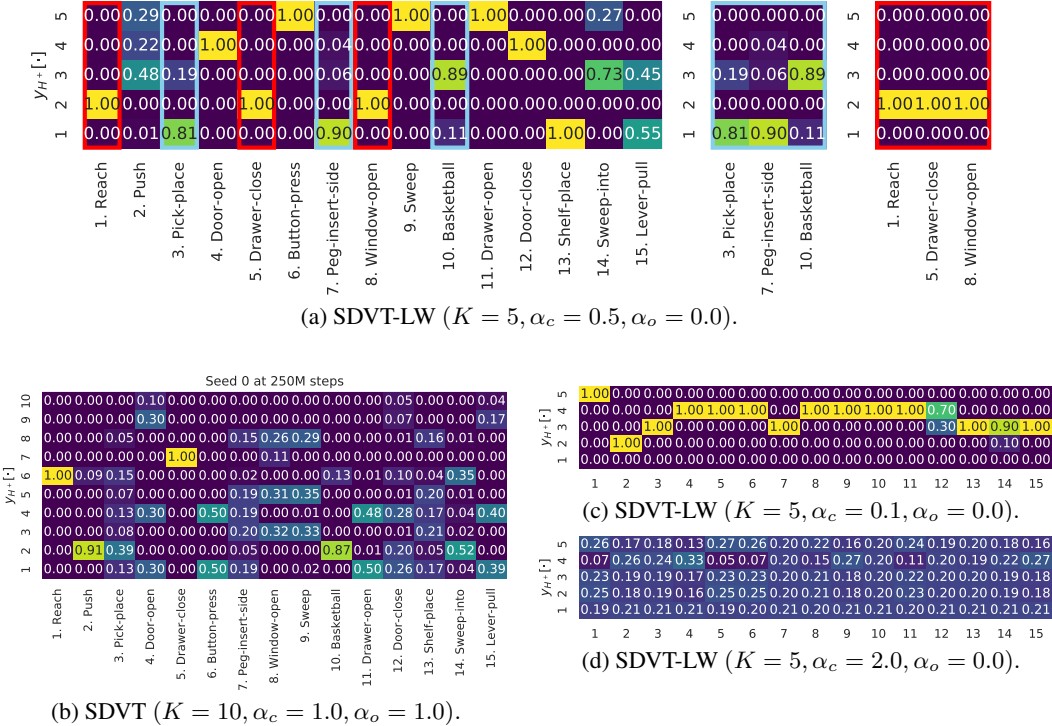

Figure 4: **Learned subtask compositions on ML-10. (a)** Default SDVT-LW **(b)** Default SDVT. **(c)** **and (d)** SDVT-LW with varying $\alpha_c$. Each column displays the terminal subtask composition ($y_{H+}$) of each task learned after 250M training steps of training, averaged across 50 parametric variants. The decompositions at different timesteps can be found in Appendix D.3. The results shown are from the first random seed, as different seeds yield distinct decompositions. Results of other seeds are provided in Appendix D.4.

returns a trajectory that solves an elementary subtask, such as placing and reaching up. We observe that the trajectories vary across different $\tilde{y}$, and similar compositions result in similar trajectories.

## 6  Conclusion

In conclusion, our proposed method has demonstrated a considerable enhancement in meta-RL with non-parametric task variability. This improvement is achieved by meta-learning shareable subtask decompositions and executing virtual training on the imaginary subtask compositions. However, it's essential to acknowledge certain potential limitations beyond the scope of this study, particularly when addressing test tasks involving entirely novel subtasks and in broader setups where the action and state spaces may also vary. Despite these limitations, we believe that expanding the realm of meta-RL to accommodate a wider range of task variability is a critical research topic. Incorporating orthogonal approaches such as using offline demonstrations or test time training techniques into our method could lead to interesting future work addressing the limitations. We are optimistic that our study lays a robust foundation for future research in this field.

## Acknowledgements

This work was supported in part by the Institute of Information & Communications Technology Planning & Evaluation (IITP) grant funded by the Korea government (MSIT) (No.2022-0-00469, Development of Core Technologies for Task-oriented Reinforcement Learning for Commercialization of Autonomous Drones, 50%) and in part by the Institute of Information & Communications Technology Planning & Evaluation (IITP) grant funded by the Korea government (MSIT) (No.2022-0-00124, Development of Artificial Intelligence Technology for Self-Improving Competency-Aware Learning Capabilities, 50%)

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

# A Pseudocode

---

**Algorithm 1** Subtask Decomposition and Virtual Training (SDVT)

---

**Initialize** Encoder $q_\phi$, decoder $p_\theta$, policy $\pi_\psi$, set of training tasks $\mathcal{M}_{\text{train}}$, virtual ratio $p_v$, GMVAE buffer $\mathcal{B}_{\text{VAE}}$, policy buffer $\mathcal{B}_{\text{pol}}$, total number of meta-episodes to train on $n_{\text{meta}}$, number of rollout episodes per meta-episode $n_{\text{roll}}$, running mean of subtask composition $\bar{y}$.
**for** meta-episode $k = 0, \ldots, n_{\text{meta}} - 1$ **do**
    Sample a training task $M_k \sim p(\mathcal{M}_{\text{train}})$
    Sample a virtual meta-episode flag $V \sim \text{Bernoulli}(p_v)$
    **if** $V = 1$ **then**
        Sample an imaginary subtask composition $\tilde{y} \sim \text{Dirichlet}(\bar{y})$
    **end if**
    Reset $h_0, y_0, \mathcal{B}_{\text{pol}}$
    **for** timestep $t = 0, \ldots, n_{\text{roll}} \times H - 1$ **do**
        **if** $t \mod H = 0$ **then**
            Reset rollout episode $s_t \sim T_{0,k}(\cdot)$
        **end if**
        Sample an action $a_t \sim \pi_\psi(\cdot|s_t, \mu_{\phi_z}(h_t, y_t), \sigma_{\phi_z}(h_t, y_t), \omega_{\phi_y}(h_t))$
        Take an environment step
            $s_{t+1} \sim T_k(\cdot|s_t, a_t)$
            $r_{t+1} \leftarrow R_k(s_t, a_t, s_{t+1})$
        **if** $V = 1$ **then**
            $\tilde{z}_t \sim \mathcal{N}(\mu_{\phi_z}(h_t, \tilde{y}), \sigma^2_{\phi_z}(h_t, \tilde{y}))$
            $\tilde{\tilde{h}}_t = p_{\theta_{\hat{h}}}(\tilde{z}_t)$
            $\tilde{r}_{t+1} \sim p_{\theta_R}(\cdot|\text{DO}[s_t, a_t, s_{t+1}], \tilde{\tilde{h}}_t)$
            Replace $r_{t+1} \leftarrow \tilde{r}_{t+1}$
        **else**
            Add the transition $(s_t, a_t, s_{t+1}, r_{t+1})$ to $\mathcal{B}_{\text{VAE}}$ $\cdots$ VAE NOT TRAINED WITH VIRTUAL DYNAMICS
        **end if**
        Add the transition $(s_t, a_t, s_{t+1}, r_{t+1})$ to $\mathcal{B}_{\text{pol}}$
        Update hidden embedding $h_{t+1} = q_{\phi_h}(\tau_{:t+1})$
        Update subtask composition $y_{t+1} \sim \text{Cat}(\omega_{\phi_y}(h_{t+1}))$
        Update the running mean of subtask composition $\bar{y} \leftarrow \text{RunningMeanUpdate}(\bar{y}, y_{t+1})$
    **end for**
    Update GMVAE $\{\phi, \theta\} \leftarrow \{\phi, \theta\} + \nabla_{\{\phi,\theta\}} \sum_{t=0}^{H^+} \text{ELBO}_t$ with samples from $\mathcal{B}_{\text{VAE}}$
    Update policy $\psi \leftarrow \psi + \nabla_\psi \mathcal{J}^+_{\text{pol}}$ with samples from $\mathcal{B}_{\text{pol}}$
    Anneal virtual ratio $p_v \leftarrow p_v + \Delta p_v$
**end for**

---

# B ELBO Derivation

The GMVAE's ELBO objective is derived as follows.

$$\mathbb{E}_{d(M_k, \tau_{:H^+})}[\log p_\theta(\tau_{:H^+})] = \mathbb{E}_{d(M_k, \tau_{:H^+})}\left[\log \mathbb{E}_{q_\phi(y_t, z_t|h_t)}\left[\frac{p_\theta(\tau_{:H^+}, y_t, z_t)}{q_\phi(y_t, z_t|h_t)}\right]\right]$$

$$\geq \mathbb{E}_{d(M_k, \tau_{:H^+})}\left[\mathbb{E}_{q_\phi(y_t, z_t|h_t)}\left[\log \frac{p_\theta(\tau_{:H^+}, y_t, z_t)}{q_\phi(y_t, z_t|h_t)}\right]\right]$$

$$= \mathbb{E}_{d(M_k, \tau_{:H^+})}\left[\mathbb{E}_{q_\phi(y_t, z_t|h_t)}\left[\log \frac{p_\theta(\tau_{:H^+}|y_t, z_t)p_\theta(z_t|y_t)p(y_t)}{q_\phi(y_t|h_t)q_\phi(z_t|h_t, y_t)}\right]\right]$$

$$= \mathbb{E}_{d(M_k, \tau_{:H^+})}\left[\mathbb{E}_{q_\phi(y_t, z_t|h_t)}\left[\log p_\theta(\tau_{:H^+}|z_t) + \log \frac{p_\theta(z_t|y_t)}{q_\phi(z_t|h_t, y_t)} + \log \frac{p(y_t)}{q_\phi(y_t|h_t)}\right]\right].$$

$$(8)$$

Eq. (8) is equivalent to the ELBO objective in Eq. (3) without weighting coefficients. We assume that the reconstruction $\tau_{:H^+}$ is conditionally independent of the subtask composition $y_t$ given $z_t$.

## C    Implementation Details

### C.1    Reference Implementations

**MAML, RL$^2$, and PEARL**   To replicate the results of MAML [13], RL$^2$ [9], and PEARL [46] as reported in the Meta-World paper [71], we utilize the exact version[3] of the Garage repository [15] without any modifications. For their hyperparameters, please refer to Appendix D.7, D.8, and D.9 of the Meta-World paper. MAML is the only baseline that has the advantage to take gradient updates during the test.

**SDVT, SD, LDM, VariBAD**   Task-inference-based methods (SDVT, SD, LDM [35], and VariBAD [74]) are adapted to the Meta-World benchmark based on the VariBAD's implementation.[4]   We begin with VariBAD's hyperparameter configuration for MuJoCo [58] Ant-goal and modify certain hyperparameters to accommodate the Meta-World benchmark (e.g., batch size, network capacity, etc.). In line with VariBAD and LDM, we train SDVT's policy using PPO [49]. The GMVAE is based on an implementation[5] that employs the Gumbel-Softmax reparameterization trick [23] when sampling the categorical subtask variable. The virtual training processes of SDVT and LDM require agent-environment interaction since they use states from the real environment. Therefore, the virtual training steps are also added when counting the total number of training steps. For a comprehensive list of SDVT hyperparameters, shared among SD, LDM, and VariBAD to ensure a fair comparison, please refer to Appendix C.3 and our source code available at `https://github.com/suyoung-lee/SDVT`.

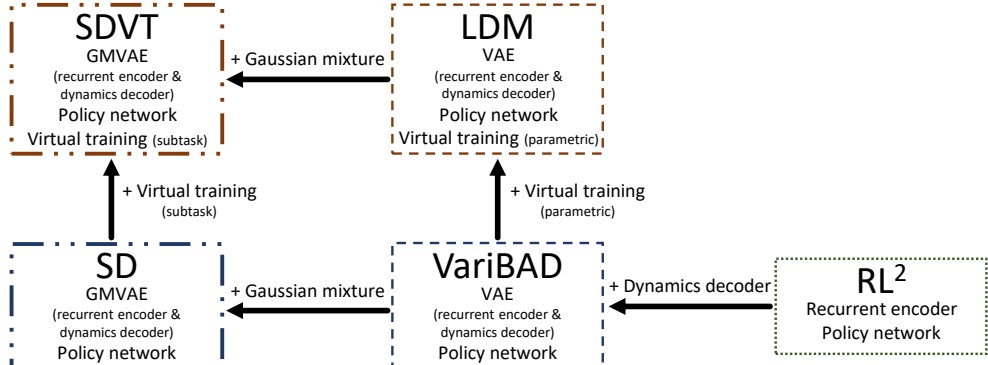

Figure 5: **Schematic overview of algorithms.** SDVT without a Gaussian mixture reduces to LDM, LDM without virtual training reduces to VariBAD, and VariBAD without a VAE decoder reduces to RL$^2$. MAML and PEARL use fully connected networks.

### C.2    Computational Complexity

Table 2: **Computational complexity.** The total wall-clock time required to generate the results for the ML-10, averaged across all eight random seeds.

|  | SDVT | SDVT-LW | SD | SD-LW | RL$^2$ | MAML | PEARL | VariBAD | LDM |
|---|---|---|---|---|---|---|---|---|---|
| Wall-clock time (hours) | 142 | 140 | 138 | 135 | 192 | 17 | 258 | 126 | 131 |

Our experiments were conducted using an Nvidia TITAN Xp. Due to the considerably larger variety of tasks compared to standard meta-RL benchmarks, non-parametric benchmarks require significantly more computational resources and time. In Table 2, we detail the total wall-clock time expended by our methods and baselines during the training and evaluation of the ML-10 benchmark. This time includes the environmental interaction time for 250M training steps, roughly 100M steps dedicated to

---

[3] `https://github.com/rlworkgroup/garage/pull/2287`
[4] `https://github.com/lmzintgraf/varibad`
[5] `https://github.com/jariasf/GMVAE`

evaluations, and the time taken to train the neural networks. Despite incorporating the GMVAE and virtual training, our method's computational demand does not substantially surpass that of VariBAD.

## C.3 Hyperparameters

Table 3: **Hyperparameters of SDVT and SD.** Hyperparameters of SDVT used for Meta-World ML-10 and ML-45 along with the notations in the manuscript and the argument names in the source code. SD shares the same hyperparameters except without those of virtual training. Different hyperparameters of our lightweight variant are denoted as (-LW).

| Category | Description | Notation | Value (ML-10, 45) | Argument Name |
|---|---|---|---|---|
| General | Rollout episode horizon | $H$ | 500 | max_episode_steps |
| | Number of rollout episodes | $n_{\mathrm{roll}}$ | 10 | max_rollouts_per_task |
| | Discount factor | $\gamma$ | 0.99 | policy_gamma |
| | Number of parallel processes | | 10 | num_processes |
| | Total Environment Steps | | $2.5e8, 4.0e8$ | num_frames |
| GMVAE | Optimizer | | Adam | optimiser_vae |
| | Learning rate | | $1e-3$ | lr_vae |
| | Subsample ELBO | | 100 | vae_subsample_elbos |
| | Subsample decodes | | 100 | vae_subsample_decodes |
| | Buffer size (meta-episodes) | $|\mathcal{B}_{\mathrm{VAE}}|$ | 1000 | size_vae_buffer |
| | ELBO categorical coefficient | $\alpha_c$ | 1.0 | cat_loss_coeff |
| | ELBO categorical coefficient (-LW) | $\alpha_c$ | 0.5 | cat_loss_coeff |
| | ELBO Gaussian coefficient | $\alpha_g$ | 1.0 | gauss_loss_coeff |
| | ELBO reward coefficient | $\alpha_R$ | 10 | rew_loss_coeff |
| | ELBO transition coefficient | $\alpha_T$ | 1000 | state_loss_coeff |
| | ELBO occupancy coefficient | $\alpha_o$ | 1.0 | occ_loss_coeff |
| | ELBO occupancy coefficient (-LW) | $\alpha_o$ | 0 | occ_loss_coeff |
| | Gumbel softmax temperature | | 1 | gumbel_temperature |
| | Number of updates per epoch | | $10, 20$ | num_vae_updates |
| | Subtask dimension | $K = N_{\mathrm{train}}$ | $10, 45$ | vae_mixture_num |
| | Subtask dimension (-LW) | $K$ | 5 | vae_mixture_num |
| | Gaussian dimension | $\dim(z)$ | $5, 10$ | latent_dim |
| | Dropout rate | $p_{\mathrm{drop}}$ | 0.7 | dropout_rate |
| | Reward reconstruction objective | | MSE | rew_pred_type |
| | State reconstruction objective | | MSE | state_pred_type |
| Policy | Algorithm | | PPO | policy |
| | Optimiser | | Adam | policy_optimiser |
| | Learning rate | | $7e-4$ | lr_policy |
| | Optimizer epsilon | | $10e-8$ | policy_eps |
| | PPO update epochs | | 5 | ppo_num_epochs |
| | Steps per policy update | $\frac{|\mathcal{B}_{\mathrm{pol}}|}{\texttt{num\_processes}}$ | 5000 | policy_num_steps |
| | Number of minibatches | | 10 | ppo_num_minibatch |
| | PPO clipping parameter | | 0.1 | ppo_clip_param |
| | GAE $\lambda$ | | 0.9 | policy_tau |
| | Initial standard deviation | | 1.0 | policy_init_std |
| | Minimum standard deviation | | 0.5 | policy_min_std |
| | Maximum standard deviation | | 1.5 | policy_max_std |
| | Entropy coefficient | | $1e-3$ | policy_entropy_coef |
| Virtual training | Initial virtual ratio | | 0.0 | virtual_ratio |
| | Virtual ratio increment per step | $\Delta p_v$ | $5e-8, 2.5e-8$ | virtual_ratio_increment |
| | ELBO dispersion coefficient | $\alpha_d$ | 10 | ext_loss_coeff |

Table 4: **Hyperparameters of VariBAD and LDM.** VariBAD and LDM employ identical hyper-parameters for the general and policy categories of SDVT and SD, as shown in Table 3. The sole distinction lies in the structural variation resulting from the utilization of GMVAE and VAE. We choose the Gaussian dimension that yielded the most favorable outcomes among 5, 10, 15, and 20.

| Category | Description | Notation | Value (ML-10, 45) | Argument Name |
|---|---|---|---|---|
| VAE | Optimizer | | Adam | `optimiser_vae` |
| | Learning rate | | $1e-3$ | `lr_vae` |
| | Subsample ELBO | | 100 | `vae_subsample_elbos` |
| | Subsample decodes | | 100 | `vae_subsample_decodes` |
| | Buffer size (meta-episodes) | $|\mathcal{B}_{\text{VAE}}|$ | 1000 | `size_vae_buffer` |
| | ELBO KL coefficient | | 0.1 | `kl_weight` |
| | ELBO reward coefficient | $\alpha_R$ | 10 | `rew_loss_coeff` |
| | ELBO transition coefficient | $\alpha_T$ | 1000 | `state_loss_coeff` |
| | Number of updates per epoch | | $10, 20$ | `num_vae_updates` |
| | Gaussian dimension | $\dim(z)$ | $5, 10$ | `latent_dim` |
| | Dropout rate (VariBAD) | $p_{\text{drop}}$ | 0.0 | `dropout_rate` |
| | Dropout rate (LDM) | $p_{\text{drop}}$ | 0.7 | `dropout_rate` |
| | Reward reconstruction objective | | MSE | `rew_pred_type` |
| | State reconstruction objective | | MSE | `state_pred_type` |

## C.4  Network Architecture

The network architectures comprising our method are described in Table 5. Prior to being input into the encoder or decoder, all state, action, and reward inputs pass through embedding networks. The values of $K = \dim(y)$ and $\dim(z)$ differ for the ML-10 and ML-45, as indicated in Table 3.

Table 5: **Network architecture.** Details of the network architecture composing the GMVAE. The numbers in the Layers column represent the dimension of the hidden layers and the output.

| Network | Notation | Architecture | Layers | Activations (last layer) |
|---|---|---|---|---|
| State embedding | | MLP | $[32]$ | (Tanh) |
| Reward embedding | | MLP | $[16]$ | (Tanh) |
| Action embedding | | MLP | $[16]$ | (Tanh) |
| Recurrent encoder | $q_{\phi_h}$ | GRU | $[256]$ | (None) |
| Categorical parameter | $\omega_{\phi_y}$ | MLP | $[512, 512, K]$ | ReLU (Softmax) |
| Gaussian parameters | $\mu_{\phi_z}, \sigma^2_{\phi_z}$ | MLP | $[512, 512, \dim(z)]$ | ReLU (None, SoftPlus) |
| Gaussian regularization parameters | $\mu_{\theta_z}, \sigma^2_{\theta_z}$ | MLP | $[\dim(z)]$ | (None, SoftPlus) |
| Latent context dispersion | $p_{\theta_{\hat{h}}}$ | MLP | $[256, 256, 256]$ | ReLU (None) |
| Reward decoder | $p_{\theta_R}$ | MLP | $[64, 64, 32, 1]$ | ReLU (None) |
| Transition decoder | $p_{\theta_T}$ | MLP | $[64, 64, 32, 40]$ | ReLU (None) |
| Policy | $\pi_\psi$ | MLP | $[256, 256, 4]$ | Tanh (Tanh) |

## C.5  Aggregation method for the Success Rate

The main results in the Meta-World paper (Table 1 and Figure 6 of Yu et al. [71]) present the average of the *maximum* success rate for each task, diverging from the raw scores in their Figure 17 and 18. For instance, if an agent achieves a 90% success rate on "Door-close" in one evaluation during training and scores 10% in all other evaluations at different times, the reported success rate for "Door-close" is 90%. The mean of these maximum scores across tasks is reported as the aggregated success rate.

This unconventional aggregation method does not accurately represent the meta-RL objective, which aims to train a single agent capable of solving multiple tasks. Meta-World calculates success rates for various tasks at distinct time points during training, even though the agent might specialize in different tasks at various stages. As a result, the agent may excel at specific tasks by chance. Consequently, evaluating the agent more frequently could yield higher *maximum* scores. We report the conventional final performance, which better reflects the meta-RL objective.

# D Detailed Experimental Results

## D.1 Performance on Individual Tasks

Table 6: **Meta-World V2 ML-10 success rate.** We report the final success rates (%) of baselines and our methods for training tasks and test tasks of the Meta-World ML-10 benchmark. All results are reported as the mean success rate $\pm$ 95% confidence interval of 8 seeds at 250M steps.

| Index. Task | SDVT | SDVT-LW | SD | SD-LW | RL$^2$ | MAML | PEARL | VariBAD | LDM |
|---|---|---|---|---|---|---|---|---|---|
| 1. Reach | 50.0±11.0 | 22.0±3.5 | 53.8±11.5 | 56.2±9.1 | 27.5±19.8 | 50.0±34.6 | **70.0±11.9** | 28.5±5.0 | 28.5±4.1 |
| 2. Push | 57.5±18.6 | 25.2±16.5 | 61.2±7.3 | **68.8±10.6** | 60.0±13.0 | 0.0±0.0 | 0.0±0.0 | 25.8±12.9 | 25.5±12.8 |
| 3. Pick-place | 47.5±9.6 | 39.0±5.4 | 52.5±7.6 | **55.0±13.9** | 42.5±21.0 | 0.0±0.0 | 0.0±0.0 | 24.8±13.3 | 27.2±13.3 |
| 4. Door-open | **100.0±0.0** | 60.5±31.9 | **100.0±0.0** | 63.8±32.5 | 93.8±9.1 | 79.5±22.1 | 10.2±11.9 | 86.8±22.7 | 62.5±33.5 |
| 5. Drawer-close | **100.0±0.0** | **100.0±0.0** | **100.0±0.0** | **100.0±0.0** | 98.8±2.3 | 99.5±1.1 | 86.8±5.6 | **100.0±0.0** | **100.0±0.0** |
| 6. Button-press | 98.8±2.3 | 98.8±1.8 | 96.2±6.9 | **100.0±0.0** | 87.5±8.3 | 93.0±8.3 | 46.0±11.2 | 98.5±0.9 | 99.5±0.9 |
| 7. Peg-insert-side | 31.2±13.2 | 24.5±10.2 | **50.0±19.9** | 36.2±11.5 | 45.0±21.6 | 0.0±0.0 | 0.0±0.0 | 19.8±14.0 | 21.8±15.3 |
| 8. Window-open | **100.0±0.0** | **100.0±0.0** | 98.8±2.3 | **100.0±0.0** | **100.0±0.0** | **100.0±0.0** | 19.2±11.5 | **100.0±0.0** | **100.0±0.0** |
| 9. Sweep | **96.2±4.8** | 72.2±29.0 | 83.8±22.2 | 83.8±15.1 | 75.0±17.3 | 0.0±0.0 | 0.0±0.0 | 60.2±32.4 | 61.2±32.9 |
| 10. Basketball | **91.2±6.4** | 78.8±9.8 | 73.8±14.3 | 91.2±8.8 | 43.8±16.6 | 0.0±0.0 | 0.0±0.0 | 37.2±26.0 | 40.5±28.9 |
| Train mean | **77.2±3.0** | 62.1±4.1 | 77.0±5.9 | 75.5±5.5 | 67.4±4.4 | 42.2±4.5 | 23.2±1.9 | 58.2±8.9 | 56.7±12.3 |
| 11. Drawer-open | **65.0±19.9** | 30.5±12.9 | 48.8±23.4 | 45.0±24.0 | 2.2±1.9 | 15.8±19.3 | 1.5±1.1 | 12.8±12.8 | 21.8±12.0 |
| 12. Door-close | 7.5±9.0 | **81.2±19.0** | 33.8±25.4 | 18.8±24.1 | 8.2±7.3 | 3.2±6.0 | 1.2±1.5 | 27.0±21.8 | 30.2±28.2 |
| 13. Shelf-place | 0.0±0.0 | **1.0±1.2** | 0.0±0.0 | 0.0±0.0 | 0.2±0.2 | 0.0±0.0 | 0.0±0.0 | 0.0±0.0 | 0.0±0.0 |
| 14. Sweep-into | **90.0±8.5** | 51.2±18.9 | 71.2±14.9 | 55.0±21.6 | 64.5±8.3 | 0.0±0.0 | 0.8±1.0 | 30.5±22.0 | 46.5±21.8 |
| 15. Lever-pull | 1.2±2.3 | 3.2±2.7 | 0.0±0.0 | **12.5±10.2** | 0.5±0.8 | 0.5±0.9 | 0.5±0.9 | 0.2±0.5 | 0.5±0.9 |
| Test mean | 32.8±3.9 | **33.4±5.0** | 30.8±7.7 | 26.2±8.7 | 15.1±2.7 | 3.9±3.7 | 0.8±0.5 | 14.1±6.1 | 19.8±6.0 |

Table 7: **Meta-World V2 ML-10 return.** We report the final return of baselines and our methods at the last rollout episode on the Meta-World ML-10 analogous to the success rate in Table 6. Results are reported as the mean $\pm$ 95% confidence intervals of 8 seeds at 250M steps.

| Index. Task | SDVT | SDVT-LW | SD | SD-LW | RL$^2$ | MAML | PEARL | VariBAD | LDM |
|---|---|---|---|---|---|---|---|---|---|
| 1. Reach | 3565±132 | 3681±115 | 3708±228 | 3806±220 | 2291±276 | **3899±560** | 3841±554 | 3829±153 | 3740±59 |
| 2. Push | **3923±339** | 3493±469 | 3636±628 | 3579±870 | 815±133 | 17±1 | 15±2 | 2609±990 | 2470±1220 |
| 3. Pick-place | **2257±239** | 2241±300 | 2131±441 | 2156±593 | 501±137 | 6±0 | 5±1 | 1395±759 | 1561±861 |
| 4. Door-open | 4441±37 | 3171±825 | **4447±106** | 3530±831 | 1249±115 | 3561±753 | 1064±254 | 4051±780 | 3249±1134 |
| 5. Drawer-close | 4813±71 | 4850±10 | 4850±13 | **4854±9** | 2193±336 | 4804±55 | 4081±314 | 4841±18 | 4837±18 |
| 6. Button-press | 3392±59 | **3437±133** | 3257±265 | 3423±209 | 1189±151 | 2574±274 | 956±300 | 3246±161 | 3252±140 |
| 7. Peg-insert-side | 1903±430 | 1919±348 | **2634±588** | 2225±592 | 682±157 | 7±0 | 7±1 | 1148±761 | 1237±825 |
| 8. Window-open | **4486±39** | 4371±77 | 4349±165 | 4398±79 | 1230±112 | 3271±732 | 784±390 | 4483±32 | 4458±42 |
| 9. Sweep | **4235±202** | 3617±682 | 3823±702 | 3557±634 | 906±223 | 79±38 | 51±14 | 2807±1260 | 2789±1333 |
| 10. Basketball | 3549±292 | **3759±307** | 3466±511 | 3726±331 | 532±116 | 7±1 | 6±2 | 2141±1119 | 2035±1256 |
| Train mean | **3656±62** | 3454±137 | 3630±241 | 3525±297 | 1159±83 | 1822±136 | 1081±77 | 3055±466 | 2963±626 |
| 11. Drawer-open | **2558±165** | 2280±271 | 2305±298 | 2310±280 | 1649±100 | 1737±270 | 1130±87 | 2336±340 | 2389±386 |
| 12. Door-close | 566±335 | **3030±857** | 1068±740 | 638±670 | 568±137 | 263±278 | 321±221 | 1121±689 | 1343±1214 |
| 13. Shelf-place | 501±77 | 550±138 | 494±102 | 434±154 | **578±70** | 0±0 | 0±0 | 211±144 | 328±204 |
| 14. Sweep-into | **2221±764** | 1455±615 | 1359±632 | 1532±721 | 490±46 | 41±3 | 48±33 | 658±358 | 1509±889 |
| 15. Lever-pull | 279±38 | 318±45 | **335±45** | 300±48 | 291±66 | 156±49 | 200±58 | 271±26 | 260±48 |
| Test mean | 1225±160 | **1527±214** | 1112±190 | 1043±234 | 715±33 | 439±78 | 340±54 | 919±143 | 1166±264 |

Table 8: **Meta-World V2 ML-45 success rate.** We report the final success rates (%) of baselines and our methods for training tasks and test tasks of the Meta-World ML-45 benchmark. All results are reported as the mean success rate $\pm$ 95% confidence interval of 8 seeds at 400M steps.

| Index. Task | SDVT | SDVT-LW | SD | SD-LW | $RL^2$ | MAML | PEARL | VariBAD | LDM |
|---|---|---|---|---|---|---|---|---|---|
| 1. Assembly | 0.0±0.0 | 0.0±0.0 | 0.0±0.0 | 0.0±0.0 | **2.2±0.4** | 0.0±0.0 | 0.0±0.1 | 0.0±0.0 | 0.0±0.0 |
| 2. Basketball | **24.8±17.7** | 3.0±3.6 | 23.8±19.2 | 24.2±13.6 | 21.0±9.8 | 0.0±0.0 | 0.0±0.0 | 19.8±13.7 | 10.2±8.9 |
| 3. Button-press-topdown | 95.0±7.6 | 94.5±0.6 | **100.0±0.0** | 98.0±1.7 | 98.8±1.0 | 78.2±21.2 | 7.5±8.6 | 99.5±0.6 | 82.5±18.7 |
| 4. Button-press-topdown-wall | 94.0±7.9 | 96.5±0.6 | 99.0±1.4 | 99.5±0.6 | 98.5±1.5 | 78.2±21.6 | 9.5±10.6 | **99.8±0.5** | 86.2±15.6 |
| 5. Button-press | 93.2±3.3 | 96.5±0.6 | 89.0±8.1 | **99.0±0.7** | 84.2±3.2 | 95.5±1.9 | 26.8±11.3 | 98.0±1.0 | 93.5±8.0 |
| 6. Button-press-wall | **86.0±11.6** | 68.0±4.0 | 80.5±8.7 | 75.5±9.4 | 67.0±4.8 | 71.2±10.6 | 20.2±8.3 | 74.5±8.3 | 56.0±11.3 |
| 7. Coffee-button | **100.0±0.0** | **100.0±0.0** | **100.0±0.0** | 97.5±2.3 | 97.5±1.6 | 92.0±4.9 | 2.8±1.3 | 78.0±14.7 | **100.0±0.0** |
| 8. Coffee-pull | 37.2±19.1 | 33.2±12.2 | 45.5±23.4 | 46.5±22.0 | **65.0±4.3** | 0.5±0.3 | 0.0±0.1 | 52.5±10.5 | 48.8±12.9 |
| 9. Coffee-push | 64.8±6.5 | 37.0±15.8 | 56.8±2.7 | 67.8±12.6 | 60.8±9.0 | 22.0±10.6 | 3.0±2.4 | 76.5±10.8 | **81.8±5.8** |
| 10. Dial-turn | **87.2±4.4** | 86.2±2.9 | 82.8±7.9 | 67.2±11.9 | 14.5±8.3 | 81.2±8.8 | 9.8±7.7 | 44.0±13.9 | 60.8±5.9 |
| 11. Disassemble | 59.5±32.0 | 32.0±14.7 | **87.8±6.2** | 74.0±10.9 | 23.2±27.1 | 0.0±0.1 | 0.2±0.2 | 44.5±21.8 | 65.5±18.5 |
| 12. Door-close | **100.0±0.0** | **100.0±0.0** | 99.5±0.6 | 97.0±3.6 | 98.2±1.2 | **100.0±0.0** | 52.2±21.2 | **100.0±0.0** | 99.2±1.0 |
| 13. Door-open | 56.2±31.2 | 56.0±24.3 | **99.8±0.5** | 64.8±26.9 | 54.0±23.8 | 6.8±11.9 | 0.5±0.3 | 98.0±1.4 | 96.2±3.6 |
| 14. Drawer-close | **100.0±0.0** | 99.0±1.2 | 97.2±2.2 | **100.0±0.0** | 98.5±0.5 | 100.0±0.1 | 96.5±2.9 | 99.2±0.7 | 96.5±4.0 |
| 15. Drawer-open | 96.5±1.1 | 83.8±16.3 | 96.0±4.8 | 87.2±13.8 | 83.5±8.4 | 0.0±0.0 | 2.5±1.3 | **100.0±0.0** | 93.2±6.6 |
| 16. Faucet-open | 99.5±0.6 | 82.5±21.0 | 99.2±1.0 | **100.0±0.0** | 86.8±9.7 | 70.5±19.1 | 28.8±11.7 | 95.5±1.7 | 98.2±1.5 |
| 17. Faucet-close | 97.0±5.5 | 77.0±18.3 | 99.2±0.7 | 97.0±2.3 | 56.8±16.8 | 52.8±22.5 | 15.0±6.3 | **99.5±0.6** | 99.2±1.0 |
| 18. Hammer | 2.0±3.7 | 0.0±0.0 | 0.0±0.0 | 1.5±1.8 | **12.8±15.4** | 6.8±5.8 | 2.2±1.3 | 0.0±0.0 | 0.0±0.0 |
| 19. Handle-press-side | 97.8±4.1 | **100.0±0.0** | **100.0±0.0** | **100.0±0.0** | 99.8±0.1 | 83.2±12.2 | 12.8±11.8 | **100.0±0.0** | 99.8±0.5 |
| 20. Handle-press | 98.5±1.1 | 99.0±1.2 | **100.0±0.0** | **100.0±0.0** | 99.8±0.1 | 72.0±20.2 | 66.8±19.8 | **100.0±0.0** | **100.0±0.0** |
| 21. Handle-pull-side | 40.0±22.3 | 5.2±5.2 | 50.5±25.2 | 39.5±24.1 | **83.2±4.3** | 13.8±15.5 | 0.8±0.7 | 41.2±23.3 | 31.8±28.5 |
| 22. Handle-pull | 0.0±0.0 | 1.0±1.2 | 35.2±31.5 | 3.8±2.6 | 0.2±0.2 | **49.0±18.9** | 0.8±0.4 | 1.0±0.7 | 1.0±1.0 |
| 23. Lever-pull | 10.0±9.8 | 16.0±4.4 | 11.5±6.8 | **23.2±16.6** | 1.0±0.8 | 3.2±3.9 | 0.0±0.0 | 20.0±11.3 | 6.0±3.9 |
| 24. Peg-insert-side | 1.0±0.7 | 0.0±0.0 | 0.8±0.7 | 0.5±0.6 | **19.5±5.8** | 0.0±0.0 | 0.0±0.0 | 0.0±0.0 | 2.2±2.7 |
| 25. Pick-place-wall | 30.0±16.7 | 29.5±9.8 | **48.5±3.7** | 41.2±11.7 | 42.5±4.0 | 6.5±6.0 | 0.0±0.0 | 35.5±4.6 | 31.2±15.9 |
| 26. Pick-out-of-hole | 26.8±16.4 | 23.2±12.0 | **53.0±7.0** | 43.8±8.4 | 5.5±2.7 | 0.0±0.0 | 0.0±0.0 | 32.0±20.8 | 34.0±11.4 |
| 27. Reach | 31.5±6.7 | 13.5±4.0 | 36.0±8.7 | 25.5±6.7 | **45.8±8.4** | 18.8±10.6 | 13.5±7.2 | 27.0±4.7 | 28.5±3.2 |
| 28. Push-back | 40.8±21.3 | 31.0±11.9 | 76.0±5.3 | 44.5±9.3 | **77.8±4.6** | 0.0±0.0 | 1.8±1.4 | 41.5±24.1 | 49.5±10.5 |
| 29. Push | 50.5±5.7 | **62.8±5.1** | 36.2±14.3 | 29.2±18.3 | 58.5±1.6 | 16.8±14.9 | 1.8±1.4 | 45.0±9.5 | 35.2±11.7 |
| 30. Pick-place | 28.2±13.5 | 26.2±6.5 | **57.5±9.1** | 36.8±3.9 | 43.5±1.8 | 4.8±5.3 | 0.0±0.0 | 34.8±3.9 | 44.0±5.2 |
| 31. Plate-slide | **75.8±5.7** | 56.5±6.9 | 58.2±10.8 | 60.2±10.4 | 20.5±14.4 | 17.0±15.2 | 9.5±10.6 | 55.2±6.8 | 57.0±7.3 |
| 32. Plate-slide-side | 81.8±0.8 | 1.0±0.7 | **97.0±3.6** | 87.0±3.2 | 70.8±5.1 | 2.0±3.6 | 0.2±0.3 | 89.5±13.6 | 59.5±24.2 |
| 33. Plate-slide-back | 82.0±6.2 | 86.0±9.7 | 83.5±4.2 | 76.5±10.8 | **90.0±3.9** | 0.0±0.0 | 0.8±0.8 | 84.0±6.1 | 80.2±7.9 |
| 34. Plate-slide-back-side | 71.8±5.6 | 59.0±4.5 | 60.8±11.2 | 73.5±10.8 | **81.5±1.9** | 5.5±5.7 | 0.5±0.6 | 73.5±2.4 | 80.8±8.0 |
| 35. Peg-unplug-side | 40.0±17.2 | 50.8±12.0 | 54.0±5.1 | 47.8±17.8 | 54.0±3.7 | 13.0±3.4 | 3.5±2.1 | **60.5±5.8** | 48.8±9.1 |
| 36. Soccer | 29.8±7.2 | 25.5±8.8 | 16.0±2.7 | 24.5±16.9 | **37.8±4.7** | 14.5±12.7 | 2.5±1.8 | 20.2±5.4 | 15.2±2.5 |
| 37. Stick-push | 0.0±0.0 | 47.5±32.9 | 0.0±0.0 | 20.5±24.6 | **88.2±3.4** | 0.0±0.1 | 0.0±0.0 | 0.0±0.0 | 0.0±0.0 |
| 38. Stick-pull | 0.0±0.0 | **15.0±10.4** | 0.0±0.0 | 12.0±14.4 | 4.5±3.6 | 0.0±0.0 | 0.0±0.0 | 0.0±0.0 | 0.0±0.0 |
| 39. Push-wall | 51.0±4.9 | 50.0±25.4 | 62.8±21.7 | 27.5±20.8 | **72.5±5.4** | 18.2±16.5 | 0.2±0.2 | 63.8±15.7 | 44.0±20.5 |
| 40. Reach-wall | 16.2±5.9 | 7.0±4.9 | 27.0±8.1 | 28.0±7.3 | **65.0±5.6** | 35.2±20.2 | 15.0±6.9 | 18.0±3.6 | 13.8±9.1 |
| 41. Shelf-place | 0.0±0.0 | 0.5±0.6 | 0.8±0.7 | 0.0±0.0 | 3.8±2.9 | 0.0±0.0 | 0.0±0.0 | **4.5±4.6** | 0.8±1.0 |
| 42. Sweep-into | 92.8±3.2 | 91.0±2.9 | 90.5±5.8 | 90.2±8.6 | 87.5±1.5 | 23.8±21.3 | 4.8±3.2 | 97.8±2.6 | **98.5±1.3** |
| 43. Sweep | 11.8±10.1 | 27.0±17.4 | 33.0±17.7 | 20.0±23.2 | **41.5±21.1** | 0.0±0.0 | 0.0±0.0 | 40.5±28.1 | 5.5±6.5 |
| 44. Window-open | 99.5±0.6 | **100.0±0.0** | **100.0±0.0** | **100.0±0.0** | 97.8±1.0 | 91.5±6.6 | 27.5±11.4 | 98.2±1.6 | 98.8±2.3 |
| 45. Window-close | **100.0±0.0** | 99.0±1.2 | **100.0±0.0** | **100.0±0.0** | 95.2±3.3 | 97.0±1.9 | 27.2±11.0 | **100.0±0.0** | 99.8±0.5 |
| Train mean | 55.6±4.2 | 50.4±4.1 | **61.0±1.7** | 56.7±1.5 | 58.0±0.4 | 32.0±1.4 | 10.3±2.4 | 57.0±1.2 | 54.1±0.9 |
| 46. Bin-picking | 0.0±0.0 | 1.5±1.8 | **2.2±1.1** | 1.2±1.0 | 1.2±0.9 | 0.0±0.0 | 0.0±0.0 | 1.8±1.1 | 0.5±0.9 |
| 47. Box-close | 0.5±0.6 | 0.0±0.0 | 0.5±0.6 | 0.0±0.0 | 0.5±0.3 | 0.0±0.0 | 0.0±0.0 | **5.5±4.6** | 0.0±0.0 |
| 48. Hand-insert | 2.2±2.3 | 3.8±3.1 | 1.0±1.2 | 1.5±1.8 | 5.2±3.0 | **23.0±15.8** | 0.0±0.0 | 1.5±1.1 | 0.8±1.0 |
| 49. Door-lock | 59.2±9.1 | **74.8±7.5** | 49.8±15.5 | 51.5±9.5 | 14.0±8.2 | 11.2±10.5 | 8.2±10.0 | 37.5±20.3 | 59.2±9.6 |
| 50. Door-unlock | **78.5±7.9** | 75.8±9.1 | 61.8±10.2 | 72.5±9.5 | 38.0±11.3 | 64.5±19.2 | 25.0±19.2 | 64.2±4.3 | 63.2±6.9 |
| Test mean | 28.1±3.2 | **31.2±1.2** | 23.0±5.1 | 25.4±2.9 | 11.8±3.2 | 19.8±6.3 | 6.7±3.3 | 22.1±3.5 | 24.8±2.9 |

Table 9: **Meta-World V2 ML-45 return.** We report the final return of baselines and our methods at the last rollout episode on the Meta-World ML-45 analogous to the success rate in Table 8. Results are reported as the mean ± 95% confidence intervals of 8 seeds at 400M steps.

| Index. Task | SDVT | SDVT-LW | SD | SD-LW | RL$^2$ | MAML | PEARL | VariBAD | LDM |
|---|---|---|---|---|---|---|---|---|---|
| 1. Assembly | 1813±485 | **2573±117** | 2414±233 | 2285±195 | 1202±19 | 268±70 | 208±9 | 2434±161 | 2572±56 |
| 2. Basketball | 1929±688 | 1443±524 | 2215±148 | **2581±129** | 489±38 | 100±90 | 6±2 | 1311±651 | 2227±236 |
| 3. Button-press -topdown | 3120±447 | 3297±328 | **3580±87** | 3343±176 | 1344±14 | 2660±633 | 543±273 | 3348±227 | 3208±354 |
| 4. Button-press -topdown-wall | 3217±384 | 3386±246 | **3614±110** | 3392±185 | 1359±18 | 2664±638 | 604±314 | 3337±279 | 3218±319 |
| 5. Button-press | **3100±270** | 2977±199 | 2833±208 | 2938±193 | 1165±29 | 2942±178 | 434±195 | 2878±74 | 2745±205 |
| 6. Button-press -wall | **3401±220** | 2916±94 | 3148±157 | 3165±92 | 1378±17 | 2589±341 | 327±183 | 3040±136 | 2991±108 |
| 7. Coffee-button | 1605±678 | 2196±430 | 2517±296 | 2420±303 | 1348±40 | **3342±157** | 80±20 | 2253±303 | 2610±271 |
| 8. Coffee-pull | 1032±307 | 916±323 | 1010±338 | **1305±395** | 572±35 | 38±7 | 21±5 | 1292±260 | 1244±220 |
| 9. Coffee-push | 1424±443 | 889±455 | 1254±137 | 1687±497 | 504±41 | 111±47 | 20±10 | 1901±413 | **2175±491** |
| 10. Dial-turn | 1821±478 | 1704±520 | **1835±414** | 1825±484 | 944±70 | 1553±254 | 449±407 | 680±130 | 1101±571 |
| 11. Disassemble | 2451±1226 | 897±350 | **3387±382** | 2840±485 | 2642±1397 | 130±58 | 150±25 | 1564±749 | 2268±688 |
| 12. Door-close | 4178±204 | **4404±22** | 4375±17 | 4251±93 | 1231±20 | 4265±158 | 2430±965 | 4181±105 | 4067±174 |
| 13. Door-open | 3086±761 | 2215±430 | **4134±151** | 2833±517 | 1312±216 | 1095±316 | 469±142 | 3741±333 | 3857±325 |
| 14. Drawer-close | 4310±274 | 4544±54 | 4392±307 | 4680±80 | 1626±25 | **4725±28** | 4539±225 | 4515±112 | 4506±206 |
| 15. Drawer-open | 3964±96 | 3624±334 | **4265±106** | 4035±221 | 2169±118 | 2259±235 | 1288±367 | 4180±114 | 3876±270 |
| 16. Faucet-open | 4572±68 | 4057±692 | 4582±81 | **4674±21** | 1925±193 | 3757±540 | 1810±497 | 4338±155 | 4479±118 |
| 17. Faucet-close | 4166±351 | 3656±471 | **4457±194** | 4300±95 | 1662±115 | 3298±561 | 1470±414 | 4267±57 | 4401±196 |
| 18. Hammer | 466±60 | 465±17 | 469±5 | 459±16 | **738±233** | 649±135 | 420±20 | 476±4 | 384±115 |
| 19. Handle-press -side | 4676±192 | 4722±14 | 4767±38 | **4797±17** | 2276±43 | 3696±522 | 652±524 | 4772±19 | 4737±51 |
| 20. Handle-press | 4601±74 | 4355±183 | 4480±147 | **4789±14** | 2239±86 | 2902±791 | 3036±958 | 4728±22 | 4747±55 |
| 21. Handle-pull -side | 853±523 | 83±30 | **1507±832** | 907±731 | 565±43 | 199±300 | 12±5 | 991±576 | 917±826 |
| 22. Handle-pull | 1469±153 | 1491±110 | **2260±666** | 1541±100 | 2048±86 | 2001±597 | 29±19 | 1549±57 | 1612±170 |
| 23. Lever-pull | 446±84 | 496±74 | 483±39 | **500±106** | 305±21 | 264±60 | 119±38 | 457±32 | 345±27 |
| 24. Peg-insert -side | 767±408 | 757±48 | **1106±186** | 1067±192 | 640±77 | 33±24 | 7±2 | 778±101 | 962±232 |
| 25. Pick-place -wall | 1647±893 | 2011±717 | **2765±64** | 2570±543 | 491±22 | 389±359 | 0±0 | 2042±501 | 1873±854 |
| 26. Pick-out -of-hole | 809±435 | 1015±365 | 1319±183 | **1385±310** | 270±26 | 20±1 | 8±1 | 1129±261 | 1247±392 |
| 27. Reach | 2682±225 | 2448±313 | 3048±238 | **3116±182** | 2273±49 | 2447±829 | 2466±1079 | 2843±147 | 3085±183 |
| 28. Push-back | 674±362 | 514±154 | **1438±156** | 1218±334 | 433±42 | 11±7 | 8±3 | 871±372 | 1404±324 |
| 29. Push | 3130±290 | 3139±655 | 2961±253 | 2581±530 | 945±32 | 634±555 | 34±13 | 3427±185 | **3524±200** |
| 30. Pick-place | 1221±650 | 1582±433 | **2145±159** | 1992±122 | 576±23 | 247±222 | 5±1 | 2034±48 | 2128±144 |
| 31. Plate-slide | **3526±161** | 2950±232 | 3022±196 | 2828±303 | 2863±1349 | 1043±789 | 561±490 | 2858±251 | 2801±231 |
| 32. Plate-slide -side | 3058±289 | 1362±37 | **3471±258** | 2761±317 | 949±20 | 322±249 | 56±35 | 3107±167 | 2351±462 |
| 33. Plate-slide -back | 3783±204 | 3790±239 | 3844±128 | 3754±272 | 1544±10 | 581±364 | 655±259 | **3900±155** | 3720±325 |
| 34. Plate-slide -back-side | 3732±91 | 3764±149 | 3755±256 | 3987±228 | 1561±7 | 676±420 | 133±56 | **4099±72** | 4094±130 |
| 35. Peg-unplug -side | 982±390 | 971±331 | 1187±111 | 1079±464 | 401±36 | 62±26 | 45±28 | **1523±142** | 1087±282 |
| 36. Soccer | 1374±169 | **1566±207** | 1311±129 | 1348±332 | 671±31 | 439±359 | 98±60 | 1399±90 | 1486±82 |
| 37. Stick-push | 47±70 | 786±533 | 11±2 | 291±333 | **955±14** | 128±212 | 6±1 | 17±2 | 10±4 |
| 38. Stick-pull | 25±23 | **905±616** | 14±3 | 600±702 | 730±46 | 39±52 | 7±1 | 19±3 | 11±4 |
| 39. Push-wall | 2547±426 | 2197±668 | **3291±421** | 2496±639 | 877±85 | 597±532 | 20±7 | 2813±716 | 2744±549 |
| 40. Reach-wall | 1871±157 | 1889±412 | 2447±462 | 2741±456 | 2303±89 | **2802±1052** | 1741±641 | 2102±128 | 1836±569 |
| 41. Shelf-place | 608±308 | 657±195 | 920±55 | **1056±165** | 623±61 | 9±8 | 0±0 | 795±186 | 725±167 |
| 42. Sweep-into | 3755±333 | 3847±400 | 3869±214 | 3584±495 | 793±67 | 623±541 | 148±82 | 3750±159 | **4151±185** |
| 43. Sweep | 1148±98 | 1433±477 | **1905±449** | 1415±623 | 872±230 | 160±82 | 36±9 | 1865±799 | 1219±325 |
| 44. Window-open | 3789±249 | 4004±190 | 4138±56 | 4121±53 | 1163±37 | 2463±1004 | 705±184 | **4177±123** | 3986±154 |
| 45. Window-close | 4201±166 | 4333±135 | 4293±209 | **4496±29** | 1152±109 | 3215±425 | 990±417 | 4346±111 | 4450±33 |
| Train mean | 2379±214 | 2294±202 | **2672±79** | 2578±64 | 1411±22 | 1388±104 | 597±121 | 2492±47 | 2515±67 |
| 46. Bin-picking | 114±44 | 113±56 | **322±70** | 140±72 | 265±73 | 37±6 | 12±5 | 127±27 | 90±29 |
| 47. Box-close | 190±49 | 128±2 | 143±5 | 137±9 | **820±159** | 215±26 | 176±40 | 243±83 | 151±10 |
| 48. Hand-insert | 299±64 | 465±101 | 386±25 | 392±64 | **467±48** | 337±188 | 6±2 | 377±61 | 398±39 |
| 49. Door-lock | 1694±302 | **1973±160** | 1359±262 | 1427±289 | 995±303 | 821±243 | 960±149 | 1358±445 | 1573±168 |
| 50. Door-unlock | **1898±158** | 1792±225 | 1721±132 | 1870±245 | 769±108 | 1880±426 | 1374±537 | 1707±324 | 1627±169 |
| Test mean | 839±74 | **894±27** | 786±69 | 793±49 | 663±100 | 658±96 | 506±122 | 762±40 | 768±63 |

### D.2 Learning Curves

In Figures 6 through 9, the mean training and test success rates are represented by solid and dashed lines respectively, with the shaded region indicating the 95% confidence interval. All results are compiled from 8 random seeds. The data corresponding to the final training steps in these plots (250M steps for ML-10 and 400M steps for ML-45) are used to present the main results in Table 1.

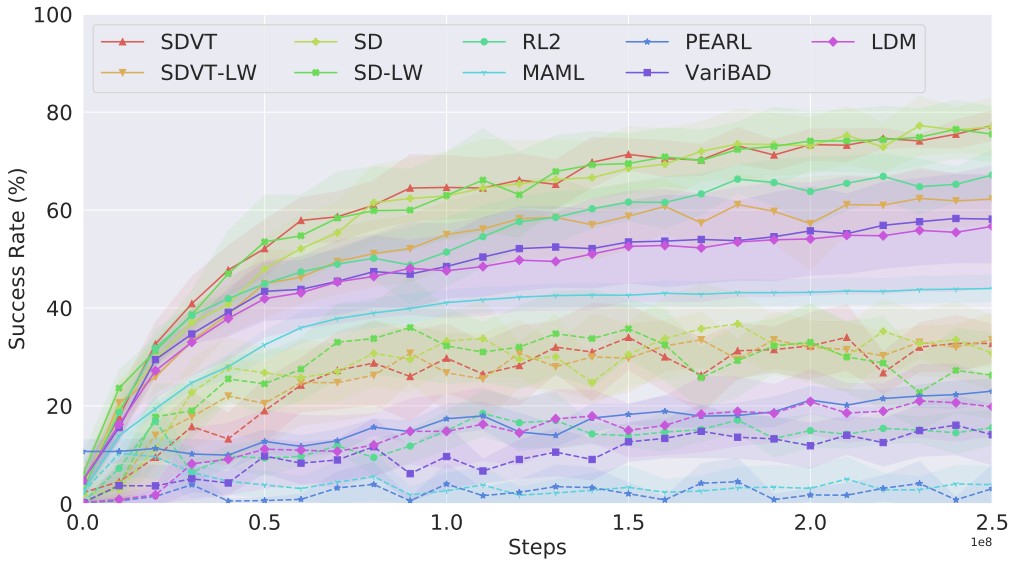

Figure 6: **Learning curve on ML-10 – success rate.**

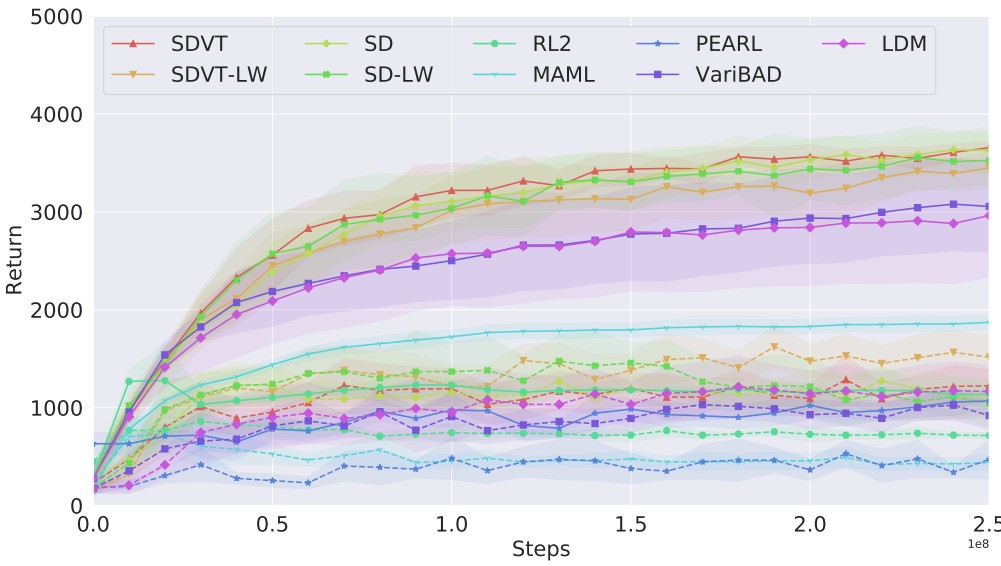

Figure 7: **Learning curve on ML-10 – return.**

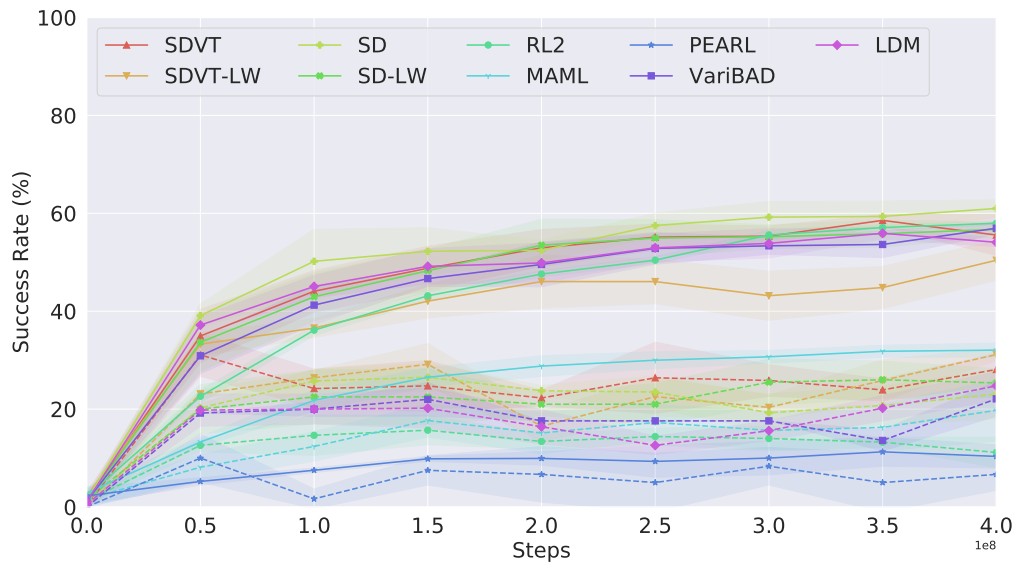

Figure 8: **Learning curve on ML-45 – success rate.**

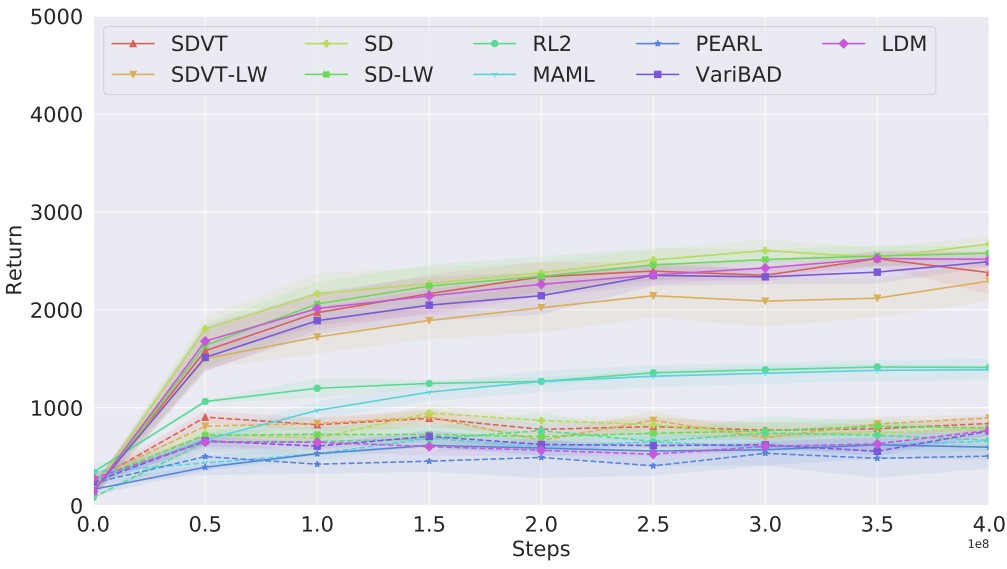

Figure 9: **Learning curve on ML-45 – return.**

## D.3 Subtask Compositions Learned over Training

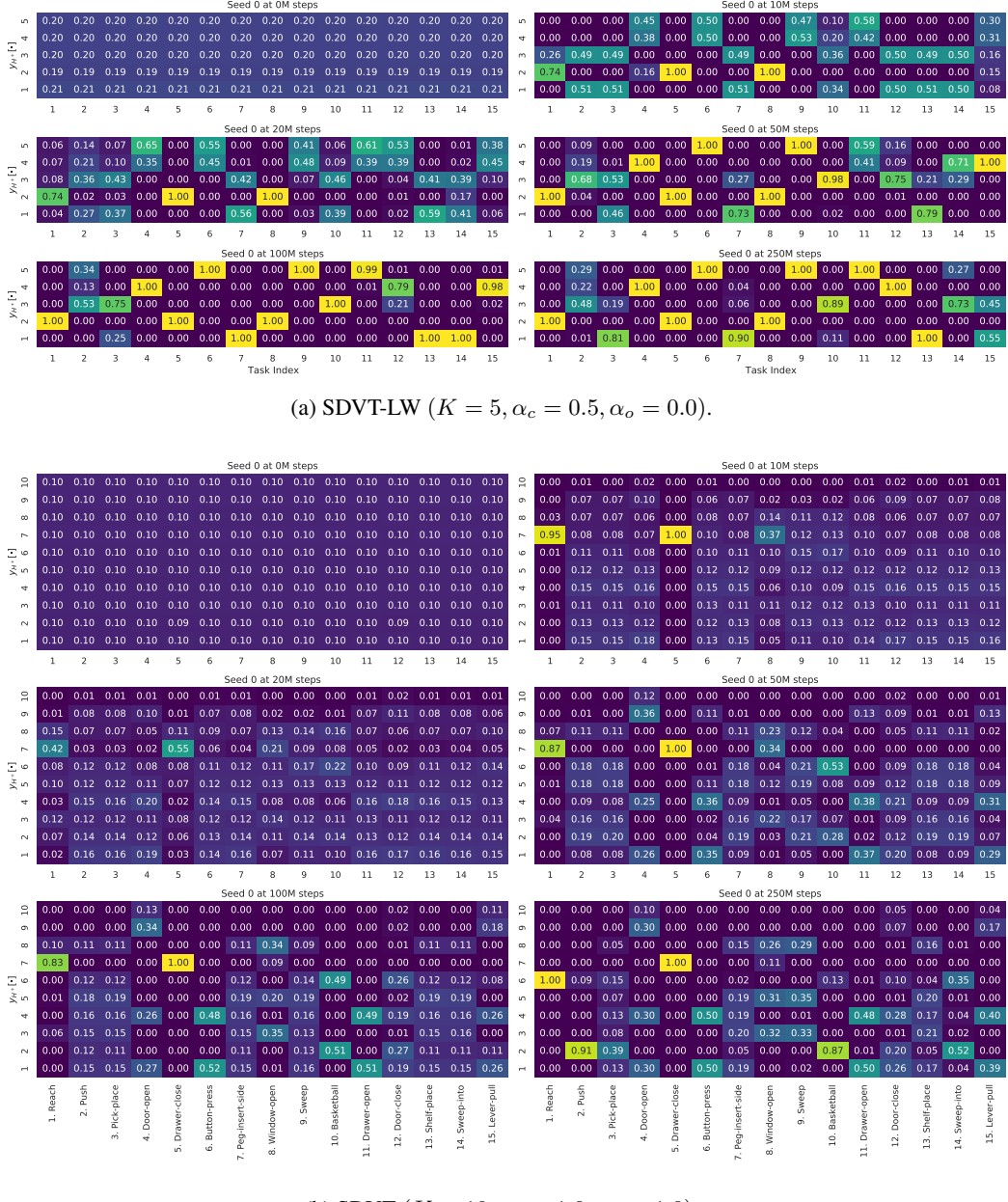

(a) SDVT-LW ($K = 5, \alpha_c = 0.5, \alpha_o = 0.0$).

(b) SDVT ($K = 10, \alpha_c = 1.0, \alpha_o = 1.0$).

Figure 10: **Subtask compositions learned over training.** We visualize the development of subtask compositions for **(a)** SDVT-LW and **(b)** SDVT during training on ML-10. Similar to Figure 4, each column represents the terminal subtask composition ($y_{H+}$) averaged across 50 parametric variants. **(a)** As training progresses, subtask compositions of (4) "Door-open" and (12) "Door-close" merge due to shared transition dynamics, which may cause the "Door-close" policy to keep the door open during testing without virtual training, underscoring the crucial role that virtual training can play in these scenarios. **(b)** With occupancy regularization, as the training progresses, the decomposition process prioritizes occupying lower indices over higher ones in the subtask compositions.

## D.4 Subtask Compositions of all Seeds

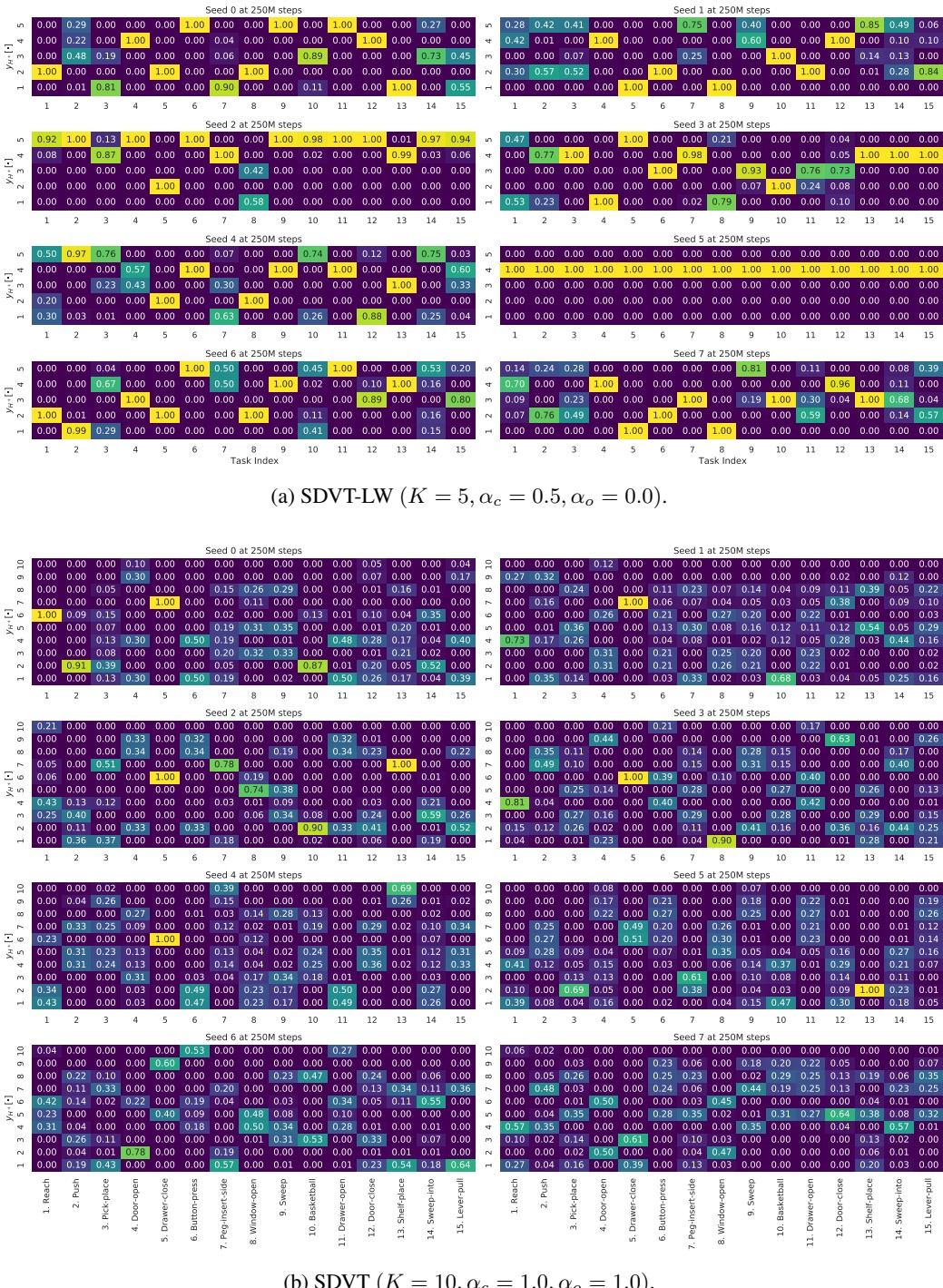

(a) SDVT-LW ($K = 5, \alpha_c = 0.5, \alpha_o = 0.0$).

(b) SDVT ($K = 10, \alpha_c = 1.0, \alpha_o = 1.0$).

Figure 11: **Subtask compositions of all seeds.** We visualize the subtask compositions of **(a)** SDVT-LW and **(b)** on ML-10 after 250M training steps for all seeds. Decomposition processes differ among random seeds due to varying initialization and sample tasks during meta-training. One subtask can be interpreted as a combination of multiple subtasks and vice versa, depending on the seed and the dimension $K$. We rarely have a collapse to a single Gaussian as in **(a)** Seed 5, which lowers the average performance. **(b)** With occupancy regularization, we find that the 10th element of the composition is rarely occupied, whereas the first element is occupied by many tasks.

# E Ablation Results

In order to confirm the source of the gain obtained from the proposed method, we conduct an extensive ablation study based on SDVT and SDVT-LW on ML-10.

## E.1 Occupancy Regularization

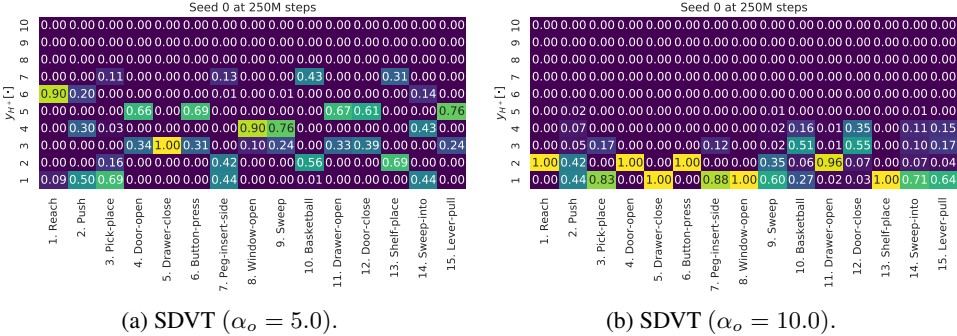

(a) SDVT ($\alpha_o = 5.0$).        (b) SDVT ($\alpha_o = 10.0$).

Figure 12: **Occupancy ablation.** We report the learned subtask compositions of SDVT ($K = 10, \alpha_c = 1.0$) for different occupancy coefficients analogous to Figure 4b.

Figures 4b and 12 indicate that our occupancy regularization effectively constrains the use of higher indices as intended. We fix the default choice of $\alpha_o = 1.0$ that we found to work well on ML-10. Our default SDVT with $\alpha_o = 1.0$ scores a test success rate of 32.8%, which scored the best among $\alpha_o \in \{0.0, 1.0, 5.0, 10.0\}$ that scored 22.8%, 25.5%, and 21.0%, respectively. Also, our default SD with $\alpha_o = 1.0$ scores a test success rate of 30.8%, which scored the best among $\alpha_o \in \{0.0, 1.0, 5.0, 10.0\}$ that scored 22.8%, 28.4%, and 11.2%, respectively.

## E.2 Dropout and Dispersion

Table 10: **Ablation results.** We report the mean success rate (%) and the difference caused by the changes made from the default SDVT-LW ($K = 5, \alpha_c = 0.5, \alpha_o = 0.0$) on ML-10 in parenthesis.

| SDVT-LW | without | | $K$ | | | $\alpha_c$ | | |
|---|---|---|---|---|---|---|---|---|
| | Dropout | Dispersion | 3 | 7 | 10 | 0.1 | 1.0 | 2.0 |
| ML-10 Train | 65.6 (+3.5) | 73.0 (+10.9) | 60.1 (-2.0) | 69.5 (+7.4) | 70.8 (+8.7) | 59.8 (-2.3) | 70.3 (+8.2) | 66.3 (+4.2) |
| ML-10 Test | 16.5 (-16.9) | 21.5 (-11.9) | 25.0 (-8.4) | 22.0 (-11.4) | 21.1 (-12.3) | 20.5 (-12.9) | 16.1 (-17.3) | 17.9 (-15.5) |

Table 10 demonstrates the significant roles played by both dropout and dispersion in generating extrapolated dynamics during virtual training, thereby enhancing the test success rate. As outlined in Appendix E.1 of the LDM paper [35], virtual training without dropout does not exhibit a significant improvement due to decoder overfitting on the state and action inputs of the training tasks. In fact, virtual training without dropout can even lead to a decrease in test performance.

However, it is important to note that these factors are not the primary contributors to the empirical gains of our method. Dropout is specifically essential for virtual training, but not necessarily for other methods that do not employ virtual training, such as VariBAD and RL2 (as discussed in Appendix E.2 of the LDM paper). We verified this in our ML-10 experiment using VariBAD with dropout. The inclusion of dropout marginally improves the training success rate of vanilla VariBAD from 58.2% to 63.0% and the test success rate from 14.1% to 16.5%. However, the performance enhancement achieved through dropout is not as substantial for VariBAD as it is for SDVT-LW's virtual training.

### E.3 Decomposition Distribution

Table 10 emphasizes the significance of hyperparameters such as subtask dimension $K$ and categorical coefficient $\alpha_c$ in determining subtask compositions. It is crucial to carefully set these hyperparameters based on the number of training tasks and their correlations. In our lightweight (LW) approach, we have selected $K = 5$ and $\alpha_c = 0.5$, which effectively distributes subtasks across the ML-10 tasks. When $\alpha_c$ is set to a small value, task classification collapses into a few subtasks, as depicted in Figure 4c. Conversely, with a large value of $\alpha_c$, the entropy of the composition is maximized, and all tasks exhibit a uniform probability distribution, as illustrated in Figure 4d. However, it is noteworthy that SDVT-LW outperforms VariBAD and LDM in test success rate, regardless of the specific values chosen for $K$, such as $K = 3, 7, 10$. To alleviate the burden of hyperparameter tuning in our LW methods, we introduce the occupancy regularization.

### E.4 Number of parameters

Table 11: **Number of parameters and success rates.** We report the number of parameters used by our methods and baselines. We demonstrate that our gain is not from the increased capacity.

| Methods | Number of Parameters | | | | ML-10 Success Rate (%) | |
| --- | --- | --- | --- | --- | --- | --- |
| | Encoder | Decoder | Policy | Sum | Train | Test |
| SDVT-LW | 1,047,455 | 174,821 | 235,401 | 1,457,677 | 62.1 | **33.4** |
| SD-LW | 1,047,455 | 25,637 | 235,401 | 1,308,493 | **75.5** | 26.2 |
| LDM | 502,580 | 25,577 | 202,249 | 730,406 | 56.7 | 19.8 |
| **LDM (matched)** | 2,144,692 | 175,593 | 267,401 | 2,587,686 | 64.2 | 22.0 |
| VariBAD | 251,290 | 25,577 | 202,249 | 479,116 | 58.2 | 14.1 |
| **VariBAD (hidden ×2)** | 894,362 | 25,577 | 663,305 | 1,583,244 | 62.4 | 12.0 |
| **VariBAD (matched)** | 1,072,346 | 175,593 | 267,401 | 1,515,340 | 67.6 | 17.2 |

Table 11 indicates that SDVT-LW employs 3 and 2 times more parameters compared to VariBAD and LDM, respectively. The encoder of our method has more parameters than VariBAD due to the added categorical layer. The decoder of SDVT-LW includes the parameters of the dispersion layers. However, we find that the model's improvement is not mainly attributed to the increased capacity. Generally, a larger capacity does not necessarily guarantee better generalization.

**VariBAD (hidden ×2)** While selecting encoder (GRU) and policy hidden sizes, we experimented with 128, 256, and 512, discovering that 256 works best for all task-inference methods (LDM, SD only, and SDVT). Notably, VariBAD (hidden ×2) with hidden dimensions of 512 possesses more parameters than SDVT but exhibits an even lower test success rate than the vanilla VariBAD.

**VariBAD (matched)** We add MLP layers with hidden sizes of [1600, 256] into the encoder and [128] into the policy. The decoder's hidden size is increased from [64, 64, 32] to [128, 256, 160] to match the capacities of VariBAD and SDVT-LW. The components of VariBAD possess slightly more parameters than SDVT-LW. LDM employs two VariBAD encoders. While the matched baselines improve, they still lag behind ours.

### E.5 Conditioning policy on belief

Table 12: **Masking contexts.** We present the success rates (%) on ML-10, achieved by ablating the contexts fed into the policy, to illustrate the effective utilization of learned contexts by the policy.

| | SDVT-LW | SDVT-LW masked $(\omega_{\phi_y})$ | SDVT-LW masked $(\mu_{\phi_z}, \sigma_{\phi_z})$ | SDVT-LW masked $(\omega_{\phi_y}, \mu_{\phi_z}, \sigma_{\phi_z})$ |
| --- | --- | --- | --- | --- |
| ML-10 Train | 62.1 | 58.6 | 36.9 | 34.8 |
| ML-10 Test | 33.4 | 33.6 | 25.8 | 24.5 |

We employ VariBAD's stop gradient architecture to avoid the instability that may arise from the concurrent training of policy and ELBO objectives. This may lead to a potential vulnerability where the policy neglects contexts. However, in Meta-World, identical observations could belong to different tasks, such as "Reach" and "Pick-place." Therefore, the agent must rely on the inferred context. To confirm this, we evaluate SDVT by masking either $(\omega_{\phi_y})$ or $(\mu_{\phi_z}, \sigma_{\phi_z})$ in the policy input.

Our findings reveal that masking the contexts severely damages the training and test success rates. Especially, masking the Gaussian parameters of $z$ is more critical than masking the categorical parameter of $y$. However, this does not suggest that $y$ is redundant, given that the continuous context $z$ is trained to include $y$'s information. Interestingly, the test success rate, with all contexts masked (i.e., solely relying on the current state $s_t$), achieves a success rate of 24.5%, surpassing all baseline scores. Although the policy does not directly employ the context learned by the GMVAE, it can still be effectively trained in the imaginary task generated by the learned GMVAE to prepare for tests with unseen compositions of subtasks.

### E.6 Smaller number of rollouts

Table 13: **Success rate on ML-10 for smaller rollouts.** We report the success rates at three different rollouts, $n_{\text{roll}} = 1, 2, 10$. Note that the results reported in our manuscript are for $n_{\text{roll}} = 10$.

| | Training Success Rate (%) | | | Test Success Rate (%) | | |
|---|---|---|---|---|---|---|
| Methods | $n_{\text{roll}} = 1$ | $n_{\text{roll}} = 2$ | $n_{\text{roll}} = 10$ | $n_{\text{roll}} = 1$ | $n_{\text{roll}} = 2$ | $n_{\text{roll}} = 10$ |
| SDVT-LW | 61.4 | 62.3 | 62.1 | **32.8** | **32.7** | **33.4** |
| SD-LW | **74.9** | **75.1** | **75.5** | 25.8 | 25.2 | 26.2 |
| VariBAD | 57.0 | 58.4 | 58.2 | 14.5 | 15.3 | 14.1 |
| LDM | 55.5 | 56.0 | 56.7 | 18.4 | 18.5 | 19.8 |

We use $n_{\text{roll}} = 10$, following the original setup of the Meta-World benchmark [69, 71]. Notably, this benchmark utilizes dense rewards, therefore the context converges rather quickly, even within the first rollout (Figure 2b). As a result, the performance across rollouts does not exhibit substantial improvement. To provide further insight, we report the success rates for smaller $n_{\text{roll}}$ in Table 13.

