# OpenReview forum: "Parameterizing Non-Parametric Meta-Reinforcement Learning Tasks via Subtask Decomposition"
_NeurIPS.cc/2023/Conference — NeurIPS 2023 poster_

### Official Review · Reviewer_PDTa · 2023-06-23

**Soundness:** 3 good
**Presentation:** 3 good
**Contribution:** 3 good
**Rating:** 6
**Confidence:** 4

**Summary:**

This paper proposes SDVT, which addresses the non-parametric task variability problem in Meta-RL. SDVT utilizes a bi-level task encoding structure, namely a categorical encoding to address non-parametric variability and a VariBad-style continuous task belief. SDVT also utilizes imagined rewards to improve generalization. SDVT outperforms baseline Meta-RL algorithms on the hard ML10 and ML45 benchmarks.

**Strengths:**

1. Dealing with multi-modal and diverse task distributions is an important problem in Meta-RL.
2. Presentation is clear, and the paper is easy to follow.
3. The methodology is sensible.
4. Experiment results are solid and convincing.

**Weaknesses:**

Overall the paper is clearly presented. I have a few concerns about why not use imagined dynamics as well as discussion with some more recent Meta-RL methods. Please refer to the Questions part for details.

**Questions:**

1. Why SDVT does not use imagined dynamics for training? Not using imagined dynamics may decease SDVT's generalization power.
2. Recent work [1] has shown very good performance on Meta-World ML10 and ML45. I'd like to see some discussions and comparisons between SDVT and the no-instruction version of [1]. Can transformers to some extend achieve the same ability to deal with non-parametric task variability? What's the pros and cons of SDVT compared to [1]?



[1] Bing, Zhenshan, et al. "Meta-Reinforcement Learning via Language Instructions." arXiv preprint arXiv:2209.04924 (2022).

**Limitations:**

The authors have discussed some limitations. I's like to know the computation burden of SDVT. Compared to baseline algorithms like PEARL, how computationally dense SDVT is?

---

> ### Author Rebuttal · Authors · 2023-08-09
>
> We appreciate the reviewer for acknowledging the importance of the problem and the methodology. We address the questions as follows.
>
> ### **Q1.** Why not Imagine State Transition Dynamics?
>
> Our current approach focuses on tasks with varying reward dynamics and doesn't extend to imagining unseen state-transition dynamics. When attempting to use an MLP state-transition decoder to create imaginary tasks, we encountered significant errors that accumulated over longer timesteps, such as in the Meta-World benchmark (5000 steps per meta-episode). Employing an advanced generative model, such as the Transformer mentioned in **Q2.**, could potentially allow for more accurate generation of long state transitions. This remains a promising avenue for future research within our RL community.
>
> ### **Q2.** Comparison to a Related Work
>
> Thank you for introducing the interesting related work, MILLION [5] that merits a reference in our paper. Our method differs significantly from MILLION in its focus and approach. While MILLION emphasizes language instructions and utilizes a Transformer encoder, our method employs a GRU encoder and specifically leverages a Gaussian mixture VAE to model the latent space of compositional tasks.
>
> The advantage of our method lies in the explicit use of mixture components in latent features, while its limitation stems from the restricted expressive power of the GRU compared to the Transformer. We chose the GRU over the Transformer because of computational considerations; the computation required for the Transformer grows quadratically with input size, while the GRU's computation grows linearly. In a benchmark like Meta-World, with inferences over very long horizons (up to 5000 steps), the GRU was more suitable.
>
> Nevertheless, we recognize that the use of a Transformer might eventually be necessary for both effective inference and the generation of virtual dynamics (as discussed in **Q1.**). Exploring ways to compress long-horizon meta-RL tasks using a Transformer could be a promising direction for future research.
>
> ### **Q3.** Computation
>
> We direct the reviewer to Appendix C.2, where the relative computational complexity is detailed. The increase in computational burden in our method is from incorporating the GMVAE over a simple VAE, as well as the generation of virtual tasks. While these additional complexities do raise the computational requirements, the demand is still substantially within the same range as those of VariBAD and LDM.

---

> > ### Comment · Reviewer_PDTa · 2023-08-11
> > **Thank you for your response**
> >
> > I thank the authors for their efforts and their response has addressed my concerns. I am currently maintaining my scoring.

---

### Official Review · Reviewer_4fbE · 2023-06-24

**Soundness:** 2 fair
**Presentation:** 2 fair
**Contribution:** 3 good
**Rating:** 5
**Confidence:** 5

**Summary:**

The paper introduces a novel context-based meta-reinforcement learning (meta-RL) methodology designed for tasks that can be decomposed into sub-tasks, such as the pick and place operation. To tackle this specific type of environment, the authors propose the adoption of a novel encoder variant capable of predicting a categorical latent variable in addition to the conventional latent vector, which follows a normal distribution. Moreover, the authors employ a technique called virtual training, where interaction with simulated environments is facilitated through the decoder, aimed at enhancing the algorithm's generalization capabilities. The authors conducted a comparative analysis of their proposed algorithm against state-of-the-art meta-RL baselines utilizing two meta-world benchmark datasets.

**Strengths:**

1. The new VAE architecture is well-motivated, well-formulated, and seems to have empirical benefits. These ideas might be helpful for researchers in the field.
2. The evaluation is done on a well-known and very challenging benchmark. The authors did a fine job comparing to SOTA meta-rl works.
3. The occupancy regularization seems like a very appealing solution to make the agent prioritize using lower indices first
4. Overall, the paper is well-written and organized.


**Weaknesses:**

1. There lacks a formulation for the general problem definition, i.e. what does a "non-parametric" task mean? What do "simple parametric variations" (line 29) mean? I think the definitions in [2] might be the missing piece. This makes it hard to understand which type of environment this work is aimed at.
2. Some design choices are not well motivated or compared to different design choices (see the questions section)
3. There is no evaluation over simple and standard meta-rl environments (like simple goal-reaching). Although this is not the focus of this work, it is important to check the performances are on par with previous works.
4. The experiment on the number of parameters in Appendix E.4 is very important, yet not complete. The authors show that even after matching the total number of parameters to SDVT, VariBAD performances are still inferior. This experiment does not prove that the gain in performance is not from the capacity of the encoder or decoder (which is very plausible since the length of the meta-episode is much longer than the one used in previous works). In order for the experiment to be complete, the authors should run LDM and VariBAD with an encoder, decoder, and policy of the same sizes as SDVT-LW.


**Questions:**

1. If I understand correctly, we can parametrize the tasks in meta-world, e.g. theta = (i, theta_l), where i is the task type number in meta-world and theta_l is the low dimensional parametric variation within each task type. So what do you mean by “non-parametric” tasks?
2. In the occupancy regularization section, in the case K* is unknown, what is the motivation for making the effective K smaller? In your appendix you showed that the regularization factor of the occupancy regularization affects the success rate of SDVT, I wonder if it only affects the virtual training part. Can you please run the same ablation for SD as well?
3. I couldn’t understand where the Gaussian mixture distribution is being modeled. You state (line 133) that the encoder is a multivariate Gaussian encoder, did you mean it is a Gaussian mixture encoder? If so, can you provide an exact formulation of this distribution? In case it’s not a mistake, and the encoder is indeed a multivariate Gaussian encoder, what is the input to the mean and variance layers of the encoder? The concatenation of h_t and y_t?
4. Is the decoder dropout you present identical to the one in LDM? If so, can you refer to it in the paper?
5. In your experiments, you used n_roll=10, did you observe an increased return between rolls? Can you provide a graph of the return of each episode within a test meta episode? Can you compare your method to VariBAD/LDM but with a lower n_roll (which is standard by previous methods)?
6. In the imaginary reward generation section, you define the way the imaginary subtask compositions are being sampled (line 205), how is this different than LDM, can you compare the two? Why did you choose a Dirichlet distribution? Is there any theoretical standing for that? Please refer to [2], where the authors used KDE, shouldn’t it perform better?
7. Why did you decide to keep only the imaginary subtask compositions constant during virtual training and not the final imaginary context (as done in LDM and in [2])?
8. In section 2.1 you formulate the problem for finite train and test tasks spaces, and in section 5.1 you state ML-10 and ML-45 have N_train=10 and N_train=45 respectively. This is a bit confusing because, to the best of my knowledge, these are actually infinite train and test task spaces (due to the parametric variations), can you clarify this?
9. Related work - while mentioning MAML, I think it’s variant MMAML [1], might be more relevant to your case. Also, I think, [2] and [3] are also relevant to your work, please consider referring to them in your related work.

[1] Vuorio et al. - Multimodal Model-Agnostic Meta-Learning via Task-Aware Modulation

[2] Rimon et al. - Meta Reinforcement Learning with Finite Training Tasks -- a Density Estimation Approach

[3] Marzari et al. - Towards Hierarchical Task Decomposition using Deep Reinforcement Learning for Pick and Place Subtasks


**Limitations:**

The authors addressed the limitations of the paper

---

> ### Author Rebuttal · Authors · 2023-08-09
>
> We appreciate the reviewer for recognizing our contribution and posing many insightful questions. We are confident that by addressing these points, our manuscript will offer a more comprehensive and clearer understanding.
>
> ### **W1.** and **Q1.** "Non-parametric" Problem Formulation
>
> We apologize for the confusion, we will revise it as our global response **G1.**
>
> ### **W3.** Simple MuJoCo Tasks
>
> We evaluated SDVT-LW on 3 MuJoCo tasks defined in Table 1 of the LDM paper[2]. We report the result in Figure R2 attached to the global response. Ours performs comparable to baselines although it's not specifically designed nor tuned for simple parametric variations. We used the same hyperparameters as LDM, except for components unique to ours, such as the GMVAE: $K=4$ for Ant-direction, and $K=2$ for Ant-goal and Half-cheetah-velocity. We referred to the results of the baselines reported in Figure 7 of the LDM paper.
>
> ### **W4.** Capacity Matching
>
> We agree that we should match the capacity component-wise for a fair comparison. We have made adjustments to VariBAD and LDM accordingly: added an MLP with hidden sizes [1600, 256] after $h\_t$ in the encoder (post-GRU output), added a final layer of size [128] to the policy, and enlarged the decoder's hidden dimensions from [64,64,32] to [128,256,160]. The matched VariBAD components have slightly more parameters than those of SDVT-LW. Table R1 in the attached PDF indicates that while the matched baselines improve, they still lag behind ours.
>
> ### **Q2.** Occupancy Regularization
>
> We address the occupancy regularization in our global response **G2.**. In addition to what is discussed there, it's important to note that the occupancy regularization is not solely intended to influence the virtual training. Rather, it compresses the effective dimension of subtasks to be generalized for the test. We run the same ablation for SD with $\alpha\_o=0.0, 5.0, 10.0$ that scored test success rates of 22.8\%, 28.4\%, and 11.2\%, respectively on ML-10. The default SD with $\alpha\_o=1.0$ scored 30.8\%.
>
> ### **Q3.** Multivariate Gaussian
>
> We apologize for any confusion from the lack of detailed explanations. To clarify, both VariBAD and LDM employ a single multivariate Gaussian encoder to capture contexts. Our approach formulates the encoder as a mixture of such multivariate Gaussian encoders. It's worth noting that there are multiple ways to configure a Gaussian mixture. Our implementation differs from those marginalizing the categorical variable $y\_{t}$, but is conceptually similar to the M2 model presented in [4]. As the reviewer rightly pointed out, the input to the mean and variance MLPs is the concatenation of $h\_t$ and $y\_t$. For a more comprehensive understanding of our formulation, we direct the reviewer to the GMVAE repository we linked in Appendix C.1. (Line 451). We intend to incorporate these clarifications into the final manuscript to enhance comprehension.
>
> ### **Q4.** Dropout
>
> Yes, we'll state that the dropout structure is exactly the same as LDM.
>
> ### **Q5.** Less Rollouts
>
> We use $n\_{\text{roll}}=10$, following the original setup of the Meta-World benchmark [1]. Notably, this benchmark utilizes dense rewards, so the context converges rather quickly, even within the first rollout. This rapid convergence was illustrated in Figure 2(b) and our supplementary page (Line 305). As a result, the performance across rollouts does not exhibit substantial improvement. To provide further insight, we report the success rates for smaller $n\_{\text{roll}}$ in Table R2 of the attached PDF.
>
> ### **Q6.** Sampling Imaginary Composition
>
> In the case of LDM, a weight vector $(\alpha^{(1)},\ldots, \alpha^{(n)})$ is sampled from a Dirichlet($\mathbf{1}$) with all-one parameters and is used to compute the weighted sum of parametric contexts from $n$ independent workers from training tasks, given by $\tilde{z}\_{t}=\sum\_{i=1}^{n} \alpha^{(i)} z^{(i)}\_{t}$. Whereas we sample the composition, $\tilde{y} \sim$ Dirichlet($\bar{y}$) with the mean $\bar{y}$ being the running mean of training task compositions.
>
> The rationale behind our choice is that $\tilde{z}\_{t}$ can be any continuous value, whereas $\tilde{y}$ must be a probability vector. Therefore, the Dirichlet distribution seemed appropriate to generate unseen but similar compositions. Also using $\bar{y}$ instead of $\mathbf{1}$ as the distribution's parameter performed more stably. However, we acknowledge that this choice does not have a robust theoretical standing, and we are open to exploring alternatives. The reviewer's suggestion to consider using other theoretically solid approaches, such as KDE, for sampling virtual contexts could be an exciting avenue for future work.
>
> ### **Q7.** Virtual Training on Imaginary Contexts
>
> Our approach is similar to LDM which also uses a time-varying mixture context, $\hat{m}\_t$ (eq.(1) of the LDM paper), corresponding to $\tilde{z}\_t$ in our case. To replicate the test, it seems intuitive to update $\tilde{z}\_t$ over time since it's a function of $h\_t$, which evolves in response to the previously generated virtual reward, $\tilde{r}\_t$. In simpler terms, maintaining $\tilde{y}$ constant is akin to preserving the $\alpha^{(i)}$s in LDM during the virtual meta-episode, yet it's important to note that the final contexts are not kept static.
>
> ### **Q8.** Finite Task Space
>
> Unlike some prior works on MuJoCo, which consider infinite parametric variations, the Meta-World benchmark operates under a finite task space. This constraint adds to the challenge of generalization but also contributes to a more stable evaluation. In the case of the ML-10 example, upon initializing the benchmark, all 10 training and 5 test tasks sample 50 parametric variants each, based on the random seed. This pool of 15$\times$50 tasks remains fixed throughout the entire training process.
>
> ### **Q9.** Related Work
>
> We thank the reviewer for additional related works that deserve mention in our paper.

---

> > ### Comment · Reviewer_4fbE · 2023-08-19
> >
> > I appreciate the additional clarification and results.
> > Regarding figure R2 in the added pdf -
> >
> > 1. What is the Mixreg baseline?
> > 2. When I suggested simple MuJoCo tasks in W3 I meant standard (in distribution) meta-rl environments (similar to those in VariBAD), not the OOD environments suggested in LDM. Can you please provide results over these?

---

> > > ### Author Response · Authors · 2023-08-20
> > > **Response to Additional Questions**
> > >
> > > We appreciate the reviewer for continuing the discussion. We would like to address the additional questions as follows.
> > >
> > > **Q10.** What is the Mixreg baseline?
> > >
> > > In Figure R2, we presented the baseline results from the LDM paper [6]. Specifically, Mixreg [7] is a technique that interpolates raw observations and rewards to produce augmented training data. LDM employed this baseline to demonstrate the benefit of generating mixtures in the latent space instead of the observation space.
> > >
> > > **Q11.** In-distribution MuJoCo results
> > >
> > > The additional data shown in Figure R2 represents the returns evaluated on out-of-distribution (OOD) test tasks. We can indeed evaluate the returns on the training tasks using the same network, aligning with the standard in-distribution meta-RL environments. We take the same approach outlined in Appendix F.3 of the LDM paper [6]. We utilize the same SDVT-LW neural network that's showcased in Figure R2, but this time we present the returns from the training tasks. The final returns for these tasks can be found in Table R3, with the baseline results sourced from Figure 16 of the LDM paper (200M/200M/50M steps for Ant-direction/Ant-goal/Half-cheetah-velocity).
> > >
> > > **Table R3.** Final returns  evaluated on training tasks.
> > >
> > > |         | Ant-direction | Ant-goal | Half-cheetah-velocity |
> > > |---------|:-------------:|:--------:|:---------------------:|
> > > | LDM [6]    |     **1177.8**    |  -228.2  |         -70.9         |
> > > | Mixreg [7] |     655.5     |  -322.3  |         -176.3        |
> > > | RL2 [8]    |     712.1     |  -267.7  |         -120.9        |
> > > | VariBAD [9] |     1030.4    |  **-138.9**  |         -125.9        |
> > > | ProMP [10]   |     401.4     |  -307.3  |          -53.0          |
> > > | E-MAML [11] |      425.0      |  -307.6  |         **-51.1**         |
> > > | PEARL [12]  |      811.3     |  -363.8  |         -58.7       |
> > > | SDVT-LW |     1155.5    |  -144.3  |         -68.1         |
> > >
> > > We wish to underscore that our main emphasis is not on standard parametric or in-distribution tasks. We added these experiments to show that our approach is equally effective on such basic tasks, comparing favorably with other baselines. We appreciate the reviewer's feedback and believe incorporating these added results will further solidify the validity of our proposed method.
> > >
> > > **References**
> > >
> > > [6] Lee and Chung. "Improving generalization in meta-RL with imaginary tasks from latent dynamics mixture." NeurIPS 2021.
> > >
> > > [7] Wang et al. "Improving generalization in reinforcement learning with mixture regularization." NeurIPS 2020.
> > >
> > > [8] Duan et al. "RL2: Fast reinforcement learning via slow reinforcement learning." arXiv preprint 2016.
> > >
> > > [9] Zintgraf et al. " Varibad: A very good method for bayes-adaptive deep RL via meta-learning." ICLR 2020.
> > >
> > > [10] Rothfuss et al. "Promp: Proximal meta-policy search." ICLR 2019.
> > >
> > > [11] Stadie et al. "The importance of sampling in meta-reinforcement learning." NeurIPS 2018.
> > >
> > > [12] Rakelly et al. "Efficient off-policy meta-reinforcement learning via probabilistic context variables." ICML 2019.

---

### Official Review · Reviewer_mp13 · 2023-07-04

**Soundness:** 3 good
**Presentation:** 3 good
**Contribution:** 3 good
**Rating:** 6
**Confidence:** 4

**Summary:**

This paper proposes an encoder architecture for meta-RL for tasks with non-parametric variation. The motivation for proposing a new encoder is that in order to achieve better generalization to a test-set of tasks with previously unseen non-parametric task variations, some specialized inductive bias can help. The proposed encoder is a Gaussian mixture model, where the idea is that the mixture components should correspond to the different subtasks. At training time, new combinations of the subtasks can then be created by sampling from the mixture model. The proposed method performs well on the meta-world benchmark.

## Acknowledgment

I have read the rebuttal and the following discussion and updated my review accordingly. The authors clarified aspects of the paper that were unclear to me. To reflect my new understanding of the paper, I raised my score from reject to weak accept.

**Strengths:**

## Originality
The proposed architectural change over the closest related method (LDM from Lee and Chung, 2021.) is straightforward but well motivated for the case of non-parametric task variation. Therefore, I would argue that the contribution is sufficiently original.

## Quality
The motivation for the method is good and the method achieves strong empirical results on a relevant benchmark of non-parametric task variation. The basic elements of quality are therefore in place.

## Clarity
The paper is clearly written.

## Significance
Generalization is an important topic in the RL community. This paper proposes an architecture that seems to produce a significant improvement in generalization on task distributions with non-parametric task variation. Therefore, this paper could be an important contribution to the generalization in RL and meta-RL communities.

**Weaknesses:**

## Presenting the contributions
The scope of the contributions of the paper is not described accurately enough. The proposed method reduces to a new architecture for the encoder of the LDM (Lee and Chung, 2021.) method. This is only stated in the experimental section. In my opinion, the simple contribution is enough for a good paper when it results in a strong empirical performance, as seems to be the case here. But right now it is too hard to determine what is novel. I would recommend adding a paragraph in the introduction that more carefully outlines the specific contribution of the paper.

## Justification of the occupancy regularization
- The method section says occupancy regularization is crucial, but doesn't explain why.
- The occupancy regularizer seems to act contrary to prior from the ELBO: the former pushes the $y_t$ toward zeros and latter pushes it toward a uniform distribution. You seem to want a $K$-hot prior parametrization, but I don't see why pushing specific dimensions of $y_t$ toward zero is needed.
- In the imaginary reward generation, you use the empirical mean over the $y$ as the parameter of the reward distribution anyway, so I don't see why the dimension regularization is needed.
- **Occupancy regularization** paragraph in the experiments section shows that the occupancy regularization does limit the use of the higher dimensions of the subtask decomposition, but none of the experiments show evidence that this is useful.
- At the minimum, the occupancy regularization should be empirically validated. If it is actually useful, then it would be optimally incorporated as a part of the prior distribution for the VAE as that seems to be its role. Right now it is kind of an ad hoc regularizer that is not motivated strongly.

## Missing ablations
- The virtual training (SD vs SDVT) ablation is good, it clearly shows the benefit from virtual training. However, the other design choices are not ablated. Specifically the effects of the occupancy regularization, decoder dropout, and latent dispersion are not demonstrated empirically.
- It is not clear why the -LW baseline is introduced. It was never motivated in the text.

## Weak analysis
- The **learned subtask compositions** paragraph has good kind of analysis about the results but figure 4 is not the easiest to interpret. Mapping the arbitrary task indices between the text and the figure to some similarity between the tasks is a challenge. Maybe you could cherry-pick interesting groupings of tasks, display them as bargraphs together, with some simple explanation of what is similar about those tasks? Then delegate the matrix of all task compositions to the appendix. Or use some other visualization of the subtask decomposition. The point is the chosen presentation is not easy to interpret.
- **Occupancy regularization** shows that the occupancy regularization does limit the use of the higher dimensions of the subtask decomposition, but none of the experiments show evidence that this is useful.
- **Generated imaginary tasks** paragraph has a good idea, but the figure is not very informative. You can see that the policy tends to do different things for the different decomposition vectors, but from the figure it is impossible to tell whether these behaviors are usefully and meaningfully different. I would recommend identifying some of the specific behaviors the different dimensions cause and then show how those can be controlled by choosing different subtask vectors.

## Minor issues
- The syntax for the dropout is potentially confusing, as it is applying the dropout directly on the states and actions. Unless $s_j$ and $a_j$ are the embeddings already, but that is equally confusing.
- Would be good to explain what is the target $h_t$ in the latent dispersion objective in the text.
- Figure 2 a and 3 a are confusingly similar. Not clear why you need to show both.
- The standard PEARL isn't really a zero-shot meta-RL method because it requires full trajectories to update the belief.

## Summary
In my opinion the experimental results are good enough for a strong paper. However, the motivation of the method and presentation are not good enough for publication in its current form. Most importantly, it is too hard to understand the specific contribution of the paper over the related work and the paper lacks ablation studies on central components of the algorithm. I would be willing to raise my score if the issues listed above are addressed in a revised version.

**Questions:**

- Why is the occupancy regularization needed? Couldn't you just pick a large $K$ and accept that some of the dimensions are non-informative?
- Why would overfitting lead specifically to low predicted rewards for the less often seen states? Why not high?
- How does the proposed method handle tasks where the subtasks need to be performed in sequence?
- How is LDM limited to parametric variation between the tasks only?
- What is the effect of the decoder dropout on the performance of SDVT?
- What is the effect of the latent dispersion on the performance of SDVT?
- What is the effect of the occupancy regularization on the performance of SDVT?

**Limitations:**

Limitations of the proposed method are not discussed at a sufficient level of detail.

---

> ### Author Rebuttal · Authors · 2023-08-09
>
> We appreciate the reviewer for recognizing the many strengths of our work. In response to the concerns raised, we outline below how we plan to improve the presentation of our work. We are excited to incorporate your valuable feedback, thereby contributing to the RL community more effectively.
>
> ### **W1.** Clarifying the Contributions
>
> We acknowledge that the introduction of multiple components in our work may have led to some ambiguity. To shed clearer light on our unique contributions, we will add a paragraph in the Method section, detailing the following aspects:
>
> Foremost, our primary contribution is the proposal of SDVT-LW. This model leverages a Gaussian mixture encoder for contexts, effectively parametrizing tasks by their subtask compositions, and carries out virtual training within the realm of these compositions.
>
> Furthermore, we propose SDVT with the occupancy regularization strategy. This generalizes SDVT-LW to adapt to more difficult conditions where there is a lack of prior knowledge of the optimal number of subtasks $K^*$. We initialize with a large value $K=N\_{\text{train}}$ and employ occupancy regularization to downscale higher dimensions, navigating to discover the most efficient number of subtasks even without prior knowledge.
>
> For the purposes of virtual training, we've adopted two methodologies that have found application in previous studies: dropout [2] and dispersion via structured VAE [3]. Their use in our work remains unchanged from their original applications. The efficacy of these components, within the context of virtual training, is demonstrated via ablations in Appendix E.
>
> ### **W2.** and **Q1.** Occupancy Regularization
>
> Please refer to our response to **W1.** and the global response **G2.**.
>
> ### **W3.** Clarification on Ablations
>
> We respectfully wish to direct the reviewer's attention to the fact that we have indeed provided ablations concerning occupancy regularization, dropout, and dispersion in Appendix E. In a brief summary, and consistent with the findings of prior research (Appendix E of LDM [2] and Appendix A of DiAMetR [3]), dropout and dispersion are indispensable elements when generating virtual tasks with unseen dynamics. Dropout and dispersion allow extrapolated generation over unseen (states, actions) and contexts, respectively. Occupancy regularization plays a crucial role in enhancing test success rates of SDVT, even when starting with a large subtask dimension. To improve accessibility and comprehension, we will incorporate a summary of the function of each component into the main manuscript, as also outlined in our response to **W1.**
>
> ### **W4.** Weak Analysis
>
> - Subtask Compositions: We have showcased all tasks as matrices to highlight the differences in distribution for various hyperparameters at a glance. We recognize, however, that this may have led to some difficulty in comparing specific related groups due to spatial constraints. We sincerely apologize for any confusion and are grateful for the reviewer's suggestion of selectively highlighting intriguing groups to emphasize commonalities more clearly. We believe this will markedly enhance the interpretability of our findings. Accordingly, we will revise the figure as Figure R1 in the PDF attached to the global response.
>
> - Generated Imaginary Tasks: We acknowledge that basis subtasks may not always align with human intuition, and it may be challenging to assert that a specific dimension corresponds to a particular action. The essence is to construct augmented imaginary tasks with various compositions, each exhibiting distinct behaviors. For example, one task might ignore an object, while another may involve picking it up. To better interpret these varied behaviors, please refer to our anonymous web pages (linked in Line 305 of the main manuscript) for the videos of generated virtual tasks that had been uploaded before the time of the manuscript submission.
>
> ### **W5.** Minor Issues
>
> - Dropout: We will provide further clarification that dropout is applied to the embeddings of states and actions, following the same method as LDM [2].
>
> - Figures 2 and 3: To clarify, Figure 2 pertains to the training tasks, while Figure 3 is designed to illustrate the generation of the imaginary reward, denoted as $\tilde{r}$ given a fixed imaginary composition $\tilde{y}$.
>
> - Use of "zero-shot": We recognize that our use of the term "zero-shot" to describe the absence of gradient updates may have led to confusion. To ensure clarity, we will revise this description in our manuscript.
>
> ### **Q2.** Overfitting to High Reward States
>
> The decoder is optimized over the distribution of states and actions that the policy visits. As training progresses, the policy tends to exploit states and actions associated with higher rewards, causing the decoder to overfit these states and actions. This phenomenon aligns with the observation in Figure 4 of the LDM paper [2], illustrated with a Gridworld example. This is precisely where dropout comes into play. It ensures that the decoder doesn't solely rely on the states and actions but also takes the context into account when predicting the reward.
>
> ### **Q3.** Encoding Sequential Subtasks
>
> In our current model, the sequence or order of subtasks is not explicitly encoded. We hypothesize that this information is implicitly captured within the continuous contexts of the model. Enhancing the model to deliberately include the sequential order of subtasks is an intriguing direction for future research and could provide additional insights.
>
> ### **Q4.** Limitations of LDM for Parametric Variants
>
> LDM produces virtual tasks through interpolations of contexts from a single Gaussian VAE. By referring to Figure 9 of the LDM paper [2], it can be inferred that such contexts capture the parametric details of the tasks (e.g., coordinates of goals). In contrast, our method conducts virtual training in the subtask space, allowing for the generation of unseen compositional tasks.

---

> > ### Comment · Reviewer_mp13 · 2023-08-11
> >
> > Thanks for the response. I'm satisfied with the answers to my criticisms and trust that the authors can improve the paper enough by carefully implementing their proposed changes. I am raising my score.

---

### Official Review · Reviewer_juhC · 2023-07-06

**Soundness:** 4 excellent
**Presentation:** 3 good
**Contribution:** 3 good
**Rating:** 7
**Confidence:** 3

**Summary:**

Meta-RL has proven successful in generalizing RL across tasks. Current methods struggle to generalize across tasks with novel sub-task decompositions. The authors introduce SVDT, a GM-VAE method that decomposes tasks into sub-tasks that help with this form of generalization. Virtual training is employed to further increase robustness to unseen sub-task compositions. Dropout, regularization, and dispersion is used to further enhance performance. On Meta-World SVDT significantly outperforms existing methods.

**Strengths:**

This paper has many strengths which I list below:

- It is very well written and clear
- Strong consistent performance (8 seeds) with respect well known baselines
- Combines strengths of related works into singular framework
- Adapts virtual training to the sub-task setting
- Strong ablations and qualitative analysis (in both main paper and appendix)
- Code is provided
- Website is very informative


**Weaknesses:**

The main weakness is that the approach has many hyperparameters and a lack of analysis as to why SVDT-LW outperforms SVDT with respect to generalization. In line 295, the authors have a hypothesis. It would be great if they could include experiments to validate this hypothesis in the paper.

The authors could also expand the offline skill learning related works section to include relevant sub-task decomposition works such as [1,2].

[1] Salter, Sasha, et al. "Mo2: Model-based offline options." Conference on Lifelong Learning Agents. PMLR, 2022.

[2] Ajay, Anurag, et al. "Opal: Offline primitive discovery for accelerating offline reinforcement learning." arXiv preprint arXiv:2010.13611 (2020).

In addition, the authors often use the term non-parametric task generalization. The generalization settings the authors consider in their paper are often called modularity/composition-based generalization (see [3]). Might be worth using these terms as well.

[3] Khetarpal, Khimya, et al. "Towards continual reinforcement learning: A review and perspectives." Journal of Artificial Intelligence Research 75 (2022): 1401-1476.

**Questions:**

Could the authors clarify why tasks that can be decomposed into sub-tasks are not considered parametric? Can’t the subtask ordering and labels be the parameters?

**Limitations:**

See other sections.

---

> ### Author Rebuttal · Authors · 2023-08-09
>
> We are grateful to the reviewer for recognizing the many strengths of our paper, and we have taken the opportunity to address the concerns raised regarding the perceived weaknesses in our work as detailed below.
>
> ### **W1.** Examination of Why SDVT-LW Outperforms SDVT
>
> What our model aims for is the decomposition of tasks into a mixture of subtasks (i.e., a multi-hot composition), facilitating generalization for test tasks that share common elementary subtasks. Table 10 in Appendix E.2 highlights the significant influence of the number of subtask types, represented by $K$, on the test success rate of SDVT-LW. Specifically, when $K=10=N\_{\text{train}}$, the test success rate decreases to 21.1\%. This decrease in success rate from the optimal $K=5$ can be attributed to the increased probability that each task is classified into a unique subtask (i.e., a one-hot composition). On the other hand, when $K$ is too small, our model fails to learn an adequately diverse spectrum of subtasks. Consequently, we need to identify an optimal value for $K$ within the range of $1<K<N\_{\text{train}}$.
>
> Our introduction of SDVT with occupancy regularization was aimed at achieving performance on par with SDVT-LW, even without prior knowledge of the optimal value of $K=5$. Even when beginning with a sufficiently large $K=N\_{\text{train}}$, SDVT is able to constrain the higher dimensions of subtask compositions, although not to exactly 5 dimensions, as shown in Figure 4(a). Since SDVT requires some training epochs to diminish the higher dimensions, it lags behind SDVT-LW, which uses 5-dimensional subtask compositions directly from the beginning. Exploring ways to outperform SDVT-LW without knowledge of the optimal $K$ or even without the number of training tasks, $N\_{\text{train}}$, presents an intriguing direction for future research. We will provide a clearer explanation in the main manuscript regarding the distinctions and the rationale for introducing both the SDVT and SDVT-LW models.
>
> ### **W2.** Relevant Works
>
> We are grateful to the reviewer for introducing relevant works that align with our research. These references will enhance the related works section of our manuscript. Exploring the integration of techniques developed within the offline skill-based setup as a part of our subtask-based methodology presents an interesting future development.
>
> ### **W3.** Confusing Term "non-parametric"
>
> We apologize for the confusion caused by the use of unclear terminology. We appreciate the reviewer for suggesting alternatives. Please refer to the global response **G1.** for more detail.

---

> > ### Comment · Reviewer_juhC · 2023-08-14
> > **Response to Rebuttal**
> >
> > I appreciate that the authors have agreed to modify certain terminologies in the paper to reduce ambiguity. I am glad that the authors will expand the related works section to include relevant offline skill-based methods. The authors have also clarified why SVDT-LW outperforms SDVT. I still believe this is a good paper and will keep my original score.

---

### Author Rebuttal · Authors · 2023-08-09

We appreciate the reviewers for dedicating their time and effort to recognize the strengths of our paper, and for offering constructive feedback that will undoubtedly enhance the presentation of our work.

Below are global responses that address concerns raised by multiple reviewers. Additionally, we have provided separate, detailed responses tailored to the comments and suggestions of each individual reviewer. We report additional results in 2 figures and 2 tables in the attached PDF file.

### **G1.** Clarification of the Term "Non-Parametric" Task Generalization

The terminology "non-parametric" in relation to task variability was first coined in the Meta-World paper [1]. Its purpose was to differentiate the task variability in Meta-World manipulation tasks from that in the standard MuJoCo tasks, which primarily involve simple parametric variations such as target goal positions, directions, and velocities.

We acknowledge, as underscored by Reviewers juhC and 4fbE, that those "non-parametric" tasks can indeed be parametrized, in terms of the subtask compositions (or task index) and the traditional parametrization of each subtask. A critical insight of our work is the recognition that such specific parametrization was challenging to achieve with prior methods that utilized a single Gaussian VAE. Our primary innovation lies in facilitating this type of parametrization through the application of a Gaussian mixture VAE, which allows for a more appropriate understanding and handling of such tasks.

In conclusion, we acknowledge that the term "non-parametric" could lead to misunderstandings, as it may imply an absence of parameters to parametrize the tasks, whereas our approach successfully parametrizes them. Therefore, in order to mitigate any confusion, we will clarify the scope of our work by referring to it as modularity/composition-based generalization, in line with the suggestions made by the reviewers.

### **G2.** Occupancy Regularization (SDVT-LW vs SDVT)

We emphasize the critical role of the number of subtasks (Line 160), denoted by $K$, as illustrated in Table 10 (Appendix E.2). Since the optimal value for $K$ may not always be accessible, we propose using occupancy regularization as a remedy in such cases.

As the reviewers suggested, we can initiate $K$ with an adequately large value such as $N\_{\textrm{train}}$. However, as outlined in the paragraph beginning from Line 160, this approach has a susceptibility: each task may simply be classified into one-hot subtask composition, rather than a mixture of subtasks. While this might enhance training performance, it becomes problematic during testing with unseen compositions of subtasks. In these circumstances, the parametrization struggles to generalize, as evidenced by the low test success rate for $K=10$ in Table 10 (Appendix E.2).

To mitigate this, we employ occupancy regularization, intentionally suppressing higher dimensions to prevent the subtask representation from being reduced to one-hot labels. Our response to **W1.** of Reviewer juhC further elaborates on this approach and may help alleviate this concern. The empirical gain of the occupancy regularization is reported in Appendix E.1. In essence, while SDVT-LW is built on the presumption of knowing the optimal $K$, SDVT extends SDVT-LW by integrating occupancy regularization, enabling it to navigate broader scenarios without specific knowledge of the ideal $K$.

**References.** The list of related works referred to in our responses is as follows

[1] Yu et al. "Meta-World: A benchmark and evaluation for multi-task and meta reinforcement learning." arXiv preprint 2021 (an updated version of CoRL 2019).

[2] Lee and Chung. "Improving generalization in meta-RL with imaginary tasks from latent dynamics mixture." NeurIPS 2021.

[3] Ajay et al. "Distributionally adaptive meta reinforcement learning." NeurIPS 2022.

[4] Kingma et al. "Semi-supervised learning with deep generative models." NeurIPS 2014.

[5] Bing et al. "Meta-reinforcement learning via language instructions." arXiv preprint 2022.

---

### Decision · Program_Chairs · 2023-09-21

**Decision:**

Accept (poster)

**Comment:**

This paper proposes a new architecture for generalization to RL tasks with non-parametric variation, where the main idea is train a Gaussian mixture VAE to meta-learn the subtask decomposition process. All the reviewers agreed that the proposed method is well-motivated and sufficiently novel for the given problem, and the results on Meta-World are strong and convincing. The authors clarified most of the reviewers' questions and added more results, which addressed some of the reviewer's concerns about unfair comparisons and weak analysis. Thus, I recommend accepting this paper.